

# End-to-End Graph Neural Networks for Real-Time Hydraulic Prediction in Stormwater Systems

Zanko Zandsalimi[1], Mehdi Taghizadeh[1], Savannah Lee Lynn[1], Jonathan L. Goodall[1], Majid Shafiee-Jood[1], and Negin Alemazkoor[1]

[1]Department of Civil and Environmental Engineering, University of Virginia, Olsson Hall, 151 Engineers Way, Charlottesville, VA 22903, USA

**Correspondence:** Negin Alemazkoor (na7fp@virginia.edu)

**Abstract.** Urban stormwater systems (SWS) play a critical role in protecting communities from pluvial flooding, ensuring public safety, and supporting resilient infrastructure planning. As climate variability intensifies and urbanization accelerates, there is a growing need for timely and accurate hydraulic predictions to support real-time control and flood mitigation strategies. While physics-based models such as SWMM provide detailed simulations of rainfall-runoff and flow routing processes, their computational demands often limit their feasibility for real-time applications. Surrogate models based on machine learning offer faster alternatives, but most rely on fully connected or grid-based architectures that struggle to capture the irregular spatial structure of drainage networks, often requiring precomputed runoff inputs and focusing only on node-level predictions. To address these limitations, we present GNN-SWS, a novel end-to-end graph neural network (GNN) surrogate model that emulates rainfall-driven hydraulic behavior across stormwater systems. The model predicts hydraulic states at both junctions and conduits directly from rainfall inputs, capturing the coupled dynamics of runoff generation and flow routing. It incorporates a spatiotemporal encoder–processor–decoder architecture with tailored message passing, autoregressive forecasting, and physics-guided constraints to improve predictive accuracy and physical consistency. Additionally, a training strategy based on the pushforward trick enhances model stability over extended prediction horizons. Applied to a real-world urban watershed, GNN-SWS demonstrates strong potential as a fast, scalable, and data-efficient alternative to traditional solvers. This framework supports key applications in urban flood risk assessment, real-time stormwater control, and the optimization of resilient infrastructure systems.

## 1 Introduction

Urban stormwater systems are essential infrastructures for managing surface runoff, mitigating flood risks, and protecting urban populations from water-related hazards (Aderyani et al., 2025). With rapid urbanization and growing climate variability—particularly the increasing frequency and intensity of extreme rainfall events—ensuring the reliability and resilience of these systems has become a pressing challenge (IPCC, 2021; Agonafir et al., 2023; Zandsalimi et al., 2024; Yavari et al., 2022).





To address these challenges, city planners and water utility operators face growing demands to optimize system performance, maintain network capacity, and prevent infrastructure failures (Taheri et al., 2025; Wang et al., 2025).

To support such tasks, physics-based hydraulic models, such as the Storm Water Management Model (SWMM), are widely used due to their ability to simulate complex urban drainage processes with high fidelity. However, these models are computationally intensive, as they rely on solving partial differential equations (Du et al., 2025; Li et al., 2024a). This computational burden poses limitations for time-sensitive applications such as real-time forecasting and risk analysis, as well as computationally intensive tasks like design optimization and uncertainty quantification (Li et al., 2022; Hesamfar et al., 2023). In

response, recent advances in surrogate modeling offer promising alternatives, aiming to retain acceptable levels of accuracy while substantially reducing computational cost (Razavi et al., 2012; Zhang et al., 2021; Garzón et al., 2022).

    In the context of hydrologic and hydraulic systems, surrogate models have been widely adopted to accelerate tasks such as urban flood simulation, inundation mapping, stormwater network optimization, and real-time forecasting (Yarveysi et al., 2023; Seyedashraf et al., 2021; Palmitessa et al., 2022; Tian et al., 2022; Fraehr et al., 2024; Roy et al., 2025). These applications

typically rely on supervised machine learning methods, including multi-layer perceptrons (MLPs), long short-term memory networks (LSTMs), convolutional neural networks (CNNs), and ensemble-based models such as random forests (Song et al., 2025; Al Mehedi et al., 2023; Ghimire et al., 2021; Karimi et al., 2019; Wang et al., 2025).

    While surrogate models such as MLPs, LSTMs, and CNNs have shown success in hydrologic and hydraulic modeling, they fall short in capturing the spatial structure inherent to urban stormwater systems. Fully connected architectures like MLPs

are vulnerable to the curse of dimensionality and treat input features independently, neglecting spatial relationships between drainage components (LeCun et al., 2015). LSTMs are effective at modeling temporal patterns but similarly lack mechanisms to encode spatial dependencies across the network. CNNs can extract spatial features from grid-based data but assume regular, Euclidean layouts, making them poorly suited for the irregular topology of urban drainage networks (Zandsalimi et al., 2025). In addition to these architectural limitations, many existing models are trained for single-step prediction and often struggle

with long-term forecasting, where error accumulation becomes a critical issue. Moreover, they typically do not incorporate physical constraints, which can lead to unrealistic outputs, particularly over extended simulation periods. These limitations hinder the ability of traditional surrogate models to capture flow interactions, ensure physical plausibility, and maintain system connectivity, thereby motivating the adoption of Graph Neural Networks (GNNs), which are designed to operate directly on graph-structured data and can explicitly model spatial dependencies.

GNNs offer a structure-aware alternative that is well-suited for modeling spatially interconnected systems such as urban stormwater networks. By operating directly on graph-structured data and leveraging message-passing and neighborhood aggregation mechanisms (Wu et al., 2020), GNNs can explicitly capture spatial dependencies between network components like junctions and conduits. In contrast, traditional models lack such message-passing mechanisms and treat input features either independently, as in MLPs and LSTMs, or based on fixed spatial proximity, as in CNNs. This enables GNNs to better represent

flow interactions and topological relationships within the system. GNNs have demonstrated strong performance across a range of network-based applications, including traffic flow forecasting (Cao et al., 2024; Ma et al., 2024), power system modeling (Taghizadeh et al., 2024; Khayambashi et al., 2024), water distribution systems (Li et al., 2024b; Kerimov et al., 2023; Xing





and Sela, 2022), and flood simulation in large-scale floodplains (Taghizadeh et al., 2025; Kazadi et al., 2024; Bentivoglio et al., 2023).

Despite the growing interest in GNN-based surrogate modeling, their application to urban stormwater systems remains limited. A few recent studies have explored variants of GNN architectures as surrogate models for hydraulic prediction, including nodal depth estimation (Garzón et al., 2024), flow routing and hydraulic simulation (Zhang et al., 2024), and short-term inflow and flow rate forecasting (Li et al., 2024a). However, these studies share a critical limitation that restricts their broader applicability: they rely on precomputed runoff inputs rather than learning directly from rainfall data. In other words, they still

require running hydraulic models to estimate runoff, which undermines the core objective of surrogate modeling, i.e., to bypass computationally expensive simulations. Additionally, they often focus primarily on node-level variables, with limited treatment of edge-level dynamics such as conduit flow and depth, and they do not fully integrate both spatial and temporal dependencies in a unified framework.

To address these limitations, we propose a novel end-to-end spatiotemporal GNN-based surrogate model for real-time hy-
draulic prediction in urban stormwater systems, namely GNN-SWS. The proposed GNN-SWS surrogate employs tailored message-passing mechanisms to fully reflect the network topology of stormwater systems. In contrast to prior approaches that depend on precomputed runoff inputs, our model directly learns the rainfall–runoff relationship, enabling it to emulate both the hydrological phase (runoff generation) and the hydraulic phase (flow routing) within a unified framework. It predicts both node-level (e.g., inflow, depth) and edge-level (e.g., flow, depth) hydraulic states, overcoming the common focus on nodal

outputs alone. To achieve this, we explicitly distinguish between heterogeneous node types—junctions that are hydrologically active (i.e., connected to subcatchments) and those that are not—by embedding subcatchment attributes only into the feature vectors of connected nodes. This design allows the model to structurally represent rainfall-runoff generation, consistent with SWMM model, and to jointly simulate nodal and conduit dynamics within the GNN architecture. To ensure physically realistic predictions and reduce the risk of non-physical artifacts, we incorporate physics-guided constraints into the learning process.

The model also adopts an autoregressive forecasting structure to support long-term prediction. To enable this, we integrate the pushforward trick (Brandstetter et al., 2022)—used here for the first time in this domain—to mitigate error accumulation during multi-step forecasting and enhance predictive stability over extended horizons.

**The primary contributions of this paper are as follows:**

1. An end-to-end spatiotemporal GNN surrogate model that jointly learns rainfall-runoff generation and hydraulic flow
routing directly from rainfall inputs.

2. A heterogeneous message-passing architecture that distinguishes node types based on subcatchment connectivity, that enables structured hydrologic representation and prediction of hydraulic states at both junction and conduit levels.

3. Integration of physics-guided constraints to improve physical consistency and flood detection accuracy.

4. An autoregressive forecasting structure that supports multi-step prediction of hydraulic states over extended time hori-
zons.



5. Application of the pushforward trick to enhance stability and accuracy in multi-step stormwater forecasting.

The remainder of this paper is structured as follows. Section 2 presents the conceptual background of GNNs. Section 3 introduces the proposed methodology, including the graph representation, training framework, and evaluation metrics. Section 4 describes the study area and data preparation. Section 5 reports and discusses the experimental results, including prediction accuracy, flood detection capability, and model limitations. Finally, Sec. **??** concludes the study and outlines directions for future research.

## 2 Background

This section presents the mathematical foundations of the proposed graph-based surrogate model for urban stormwater systems. The discussion begins with the physical formulation of the Storm Water Management Model (SWMM). It then outlines the core principles of GNNs, which model spatial dependencies by leveraging graph-structured representations of infrastructure. The graph network (GN) block is introduced as the primary computational unit for information propagation through the network. Finally, the encoder–processor–decoder architecture is described as the structural backbone of the surrogate model.

### 2.1 The Storm Water Management Model

SWMM is a widely adopted rainfall–runoff modeling framework developed for simulating hydrologic and hydraulic processes in urban drainage systems. It incorporates both a hydrological runoff module and a one-dimensional hydrodynamic module to simulate the complete rainfall-to-runoff transformation and its subsequent conveyance through stormwater infrastructure Rossman et al. (2022). This dual structure enables SWMM to represent the spatial and temporal dynamics of urban flooding with considerable physical fidelity.

The hydrological component simulates surface runoff generation based on precipitation inputs, land use, slope, and soil characteristics. Surface runoff is computed on a subcatchment-by-subcatchment basis using a nonlinear reservoir approach. In this study, infiltration losses are modeled using the Horton method (Horton, 1933), which empirically describes the exponential decay of infiltration capacity over time.

For the hydraulic component, flow routing is performed using the dynamic wave method, which applies the full one-dimensional Saint–Venant equations. This approach accounts for gradually varied, unsteady open-channel flow, allowing for the simulation of complex hydraulic behaviors such as flow reversal, pressurization, surcharging, and backwater effects. When dynamic wave routing is enabled, SWMM also allows for flow accumulation at junction nodes, with excess volume temporarily stored and released during subsequent time steps.

The governing equations for the hydraulic component are based on conservation principles and are expressed as follows:

$$\frac{\partial A}{\partial t} + \frac{\partial Q}{\partial x} = 0 \quad \text{(Continuity)} \tag{1}$$

$$\frac{\partial Q}{\partial t} + \frac{\partial (Q^2/A)}{\partial x} + gA\frac{\partial H}{\partial x} + gAS_f = 0 \quad \text{(Momentum)} \tag{2}$$





where $A$ is the flow cross-sectional area, $Q$ is the flow rate, $H$ is the hydraulic head, $S_f$ is the friction slope, and $g$ is the gravitational acceleration. These equations govern the dynamic response of the stormwater system and form the physical foundation upon which the proposed surrogate model is developed.

## 2.2 Graph Neural Networks (GNNs)

GNNs provide a highly suitable modeling framework for urban stormwater systems due to their ability to naturally represent and learn from network-structured data. In these systems, infrastructure elements such as junctions, subcatchments, and conduits can be intuitively represented as nodes and edges within a graph. This alignment between physical layout and graph topology enables GNNs to capture both local hydraulic interactions and broader network-wide dynamics (Garzón et al., 2024; Zhang et al., 2024).

The core mechanism behind GNNs is message passing, where each node's representation is iteratively updated by aggregating information from its neighboring nodes and connecting edges. Through this process, GNNs learn spatial dependencies across the network and can model how perturbations, such as rainfall inputs or upstream flows, propagate through the system Taghizadeh et al. (2025).

### 2.2.1 Graph neural computation block (GN block)

The core component of the GNN's message passing mechanism is the graph network block. Consider the stormwater system to be represented as a directed graph $G = (V, E)$, where $V$ is the set of nodes (e.g., junctions, outfalls) and $E$ is the set of edges (e.g., pipes, conduits). Each node $i \in V$ and edge $(i, j) \in E$ is associated with a raw feature vector $\mathbf{x}_i$ and $\mathbf{x}_{ij}$, respectively. These features are first encoded using multi-layer perceptrons (MLPs) to produce the initial hidden representations.

The encoded features ($\mathbf{x}_i$ and $\mathbf{x}_{ij}$) are then updated iteratively through a message-passing process consisting of three steps:
edge update, message aggregation, and node update.

First, the edge features are updated by incorporating information from the connected nodes:

$$\mathbf{h}'_{ij} = \psi_e \left( \mathbf{h}_{ij} \oplus \mathbf{h}_i \oplus \mathbf{h}_j \right) \tag{3}$$

where $\psi_e$ is an MLP and $\oplus$ denotes vector concatenation. This operation enables edge features to capture the relational context of their associated nodes.

Next, updated edge features are aggregated at each node to generate a message representation:

$$\mathbf{m}_i = \sum_{j \in \mathcal{N}(i)} \mathbf{h}'_{ij} \tag{4}$$

where $\mathcal{N}(i)$ is the set of neighboring nodes connected to node $i$. These messages encode local spatial dependencies and interactions from surrounding edges.

Finally, node features are updated using the original hidden node features and the aggregated message:

$$\mathbf{h}'_i = \psi_v \left( \mathbf{h}_i \oplus \mathbf{m}_i \right) \tag{5}$$





where $\psi_v$ is another MLP. This update step allows each node to integrate information from its neighborhood, progressively expanding its receptive field over multiple message-passing iterations.

This iterative process is repeated for $m$ layers, enabling the model to learn spatial interactions across the stormwater system graph. The final refined node and edge features ($\mathbf{h}_i'^{(m)}$ and $\mathbf{h}_{ij}'^{(m)}$) are then passed to the decoder to generate the hydraulic predictions.

### 2.2.2 Encoder–processor–decoder model

The GNN-SWS model embeds GN blocks in an encoder–processor–decoder architecture tailored to the graph-structured nature of urban stormwater systems. As illustrated in Fig. 1, this framework comprises three main components: the encoder, the processor, and the decoder.

The **encoder** transforms the raw input features of nodes and edges, denoted by $\mathbf{x}_i$ and $\mathbf{x}_{ij}$, into latent representations $\mathbf{h}_i$ and $\mathbf{h}_{ij}$, respectively. These encoded features provide a consistent initialization for message passing by embedding structural and contextual information into a shared hidden space.

The **processor** consists of multiple GN blocks, each responsible for iteratively refining the node and edge features. During each iteration, edge features are updated by aggregating node-level context, and node features are subsequently updated using incoming messages derived from adjacent edges. This process allows the model to propagate and integrate spatial dependencies across the drainage network through $m$ rounds of message passing.

The **decoder** maps the final node and edge embeddings to the desired hydraulic outputs. Specifically, node features are decoded to predict variables such as inflow and water depth at junctions, while edge features are used to predict flow and depth in conduits.

This encoder–processor–decoder configuration enables the model to learn complex spatial interactions while preserving the graph structure of the stormwater network. For multi-step forecasting, the model is applied recursively to generate predictions over successive future time steps.

## 3 Material and methods

This section details the graph representation of the stormwater system, the design of the loss functions for training, and the implementation of the pushforward trick to promote temporal stability during multi-step predictions.

### 3.1 Graph Representation

In the stormwater drainage network, we classify nodes into two types to distinguish their physical roles within the system. Type *a* nodes are directly connected to subcatchments and therefore receive rainfall-driven inflow contributions. Type *b* nodes, including outfalls and intermediate junctions, do not have an associated subcatchment but are essential for routing flow through the network. A detailed illustration of the input configuration, node/edge types, and prediction targets is provided in Fig. 2.



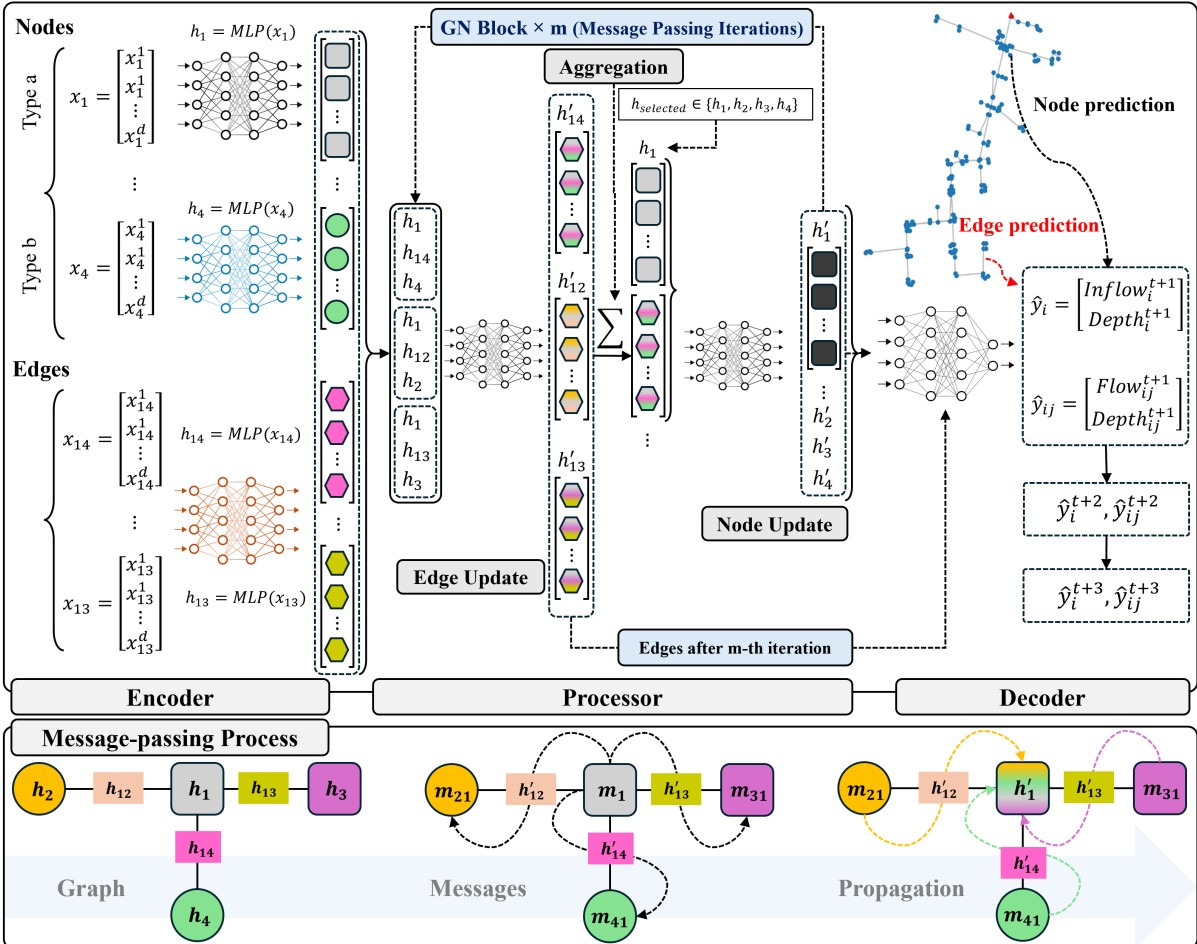

**Figure 1.** Encoder–processor–decoder architecture for rainfall-driven hydraulic prediction in stormwater systems. The processor module performs message passing through a stack of GN blocks (Graph Network blocks), each consisting of edge updates, message aggregation, and node updates.

**Static node features** include spatial and topographic information such as $x$- and $y$-coordinates, elevation, and maximum depth. For Type $a$ nodes, additional subcatchment characteristics are included, such as drainage area, slope, percentage imperviousness, and soil coverage metrics (e.g., pervious and impervious fractions).

**Dynamic node features** vary over time and reflect the evolving hydraulic state at each node. These features include rainfall intensity, accumulated rainfall, water depth, and inflow. The model receives a sequence of these features over a fixed historical window of three time steps, enabling it to capture temporal dependencies in the system's response and support multi-step autoregressive prediction.

**Dynamic and static edge features** are similarly characterized. Static edge features include geometric and hydraulic parameters such as conduit length, cross-sectional area, roughness coefficient, and slope. Dynamic edge features include time-varying







**Figure 2.** Spatiotemporal graph representation of the stormwater system used in the proposed GNN-SWS. The network is modeled as a directed graph comprising two node types: Type *a* (junctions connected to subcatchments) and Type *b* (junctions without subcatchments), along with directed edges representing conduits. Each node and edge is associated with a set of static features and a sequence of dynamic hydraulic variables over a fixed historical time window. The model uses three previous time steps as input and autoregressively forecasts hydraulic states over the next three time steps, including node-level variables (inflow, depth) and edge-level variables (flow, depth). Note that during inference, the model's own predictions from earlier steps (e.g., inflow and depth at t+1) are recursively used as inputs for later predictions (e.g., t+2 and t+3).

values of flow and depth, as well as derived attributes such as differences in water depth and inflow between upstream and downstream nodes:

$$\Delta \text{depth}_{ij} = \text{depth}_i - \text{depth}_j, \quad \Delta \text{inflow}_{ij} = \text{inflow}_i - \text{inflow}_j \tag{6}$$





These differences facilitate directional message passing in GNN, enhancing its ability to learn from localized hydraulic gradients. All static and dynamic input features, except for spatial coordinates, are globally normalized across the training 195 dataset to remove scale disparities and stabilize learning.

The overall spatiotemporal representation is constructed by combining static features with previous sequences of dynamic features across all nodes and edges. The model operates in an autoregressive forecasting mode, where predictions are generated for multiple future time steps $[t+1,\ldots,t+k]$, conditioned on dynamic inputs from previous steps $[t-n+1,\ldots,t]$. During training, these inputs and targets are based on observed data. However, during inference, the model no longer has access to 200 ground-truth values instead recursively uses its own predictions (e.g., inflow and depth at $t+1$) as inputs for subsequent steps (e.g., $t+2, t+3$). As illustrated in Fig. 3a, the model adopts a residual update strategy throughout both training and inference, where it predicts the temporal changes relative to the previous time step at each prediction step. Node-level targets include water depth and inflow, while edge-level targets consist of conduit flow and depth. This architecture enables the model to capture both spatial dependencies and temporal evolution in the hydraulic behavior of urban stormwater systems.

## 3.2 Pushforward trick for stability

In autoregressive forecasting, the model predicts future hydraulic states by recursively using its own past predictions as inputs. While this setup enables multi-step forecasting, it also introduces a key challenge known as *distribution shift*—a divergence between the input distributions observed during training and those encountered during inference. During training, the model typically receives clean, ground-truth inputs at each time step. In contrast, during inference, the model must rely on its own pre-210 vious outputs, which are inherently imperfect. As small prediction errors accumulate, they can push the model into unfamiliar regions of the input space, potentially leading to instability and degraded performance over long forecasting horizons.

To address this issue, we adopt the *pushforward trick* (Brandstetter et al., 2022), a training strategy designed to improve stability under autoregressive conditions. Instead of always conditioning on the true state of the system, the model is trained to use its own past predictions as input. This exposes the model to realistic rollout behavior and teaches it to maintain performance 215 despite deviations from the true state.

This strategy is illustrated in Fig. 4, which compares standard training with autoregressive rollout under the pushforward trick.

Let $\mathcal{F}$ denote the GNN-SWS prediction operator that maps the current system state to its next-step forecast. The model's prediction at time $t$, denoted by $\hat{\mathbf{y}}^{(t)}$, can be interpreted as a perturbed version of the true system state:

$$\hat{\mathbf{y}}^{(t)} \approx \mathbf{y}^{(t)} + \varepsilon, \tag{7}$$

where $\varepsilon$ represents the implicit prediction error. Hence, during training rollouts, the model already receives inputs that differ from the ground truth, mimicking inference-time conditions (Fig. 4b).

The next-step prediction is defined by applying the forecasting operator:

$$\hat{\mathbf{y}}^{(t+1)} = \mathcal{F}(\hat{\mathbf{y}}^{(t)}), \tag{8}$$





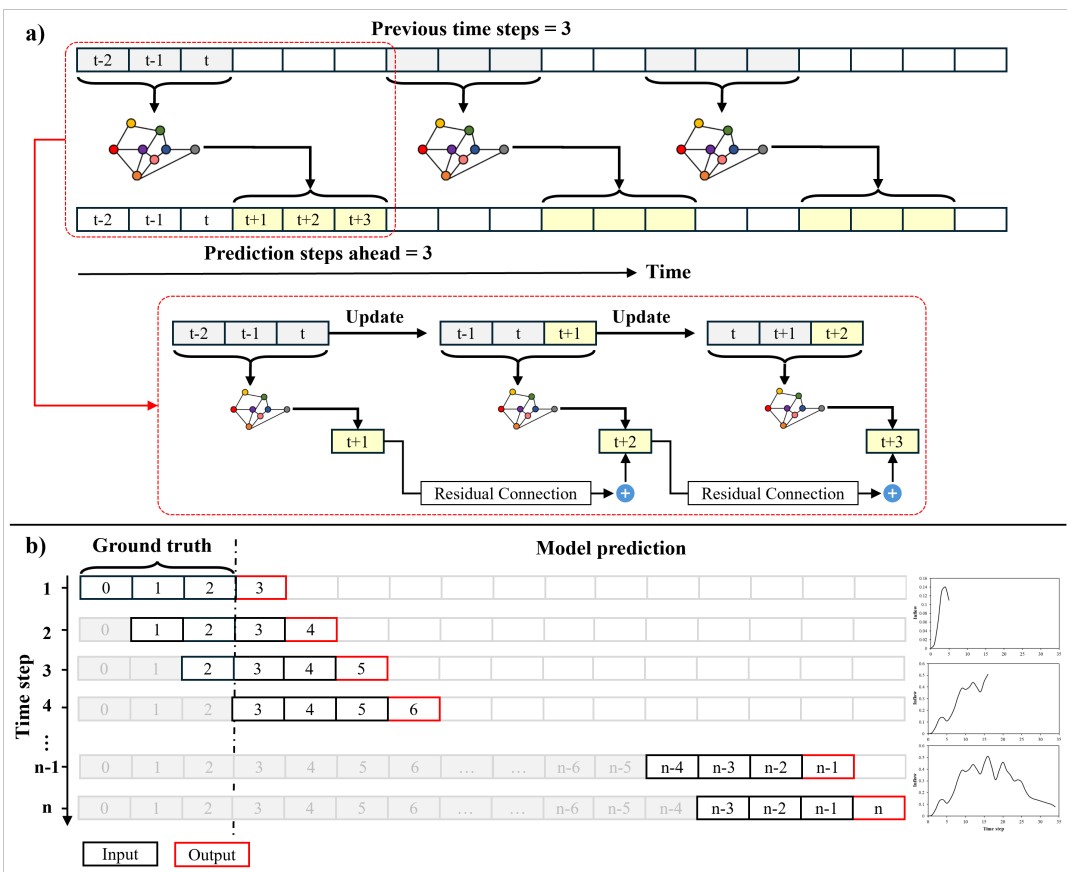

**Figure 3.** Training and testing procedure of the GNN-SWS model. **(a)** Training phase: The model uses $n$ previous time steps to predict up to $n$ steps ahead. **(b)** Rollout phase: During testing, predictions are recursively fed back into the model to generate long-term forecasts over the test dataset.

To penalize trajectory drift introduced by autoregressive inputs, we define a stability loss that compares the model's predictions to the ground truth over a fixed prediction horizon:

$$\mathcal{L}_{\text{stability}} = \frac{1}{N_t} \sum_{t=1}^{N_t} \sum_{v \in V} \left\| \mathcal{F}(\hat{\mathbf{y}}^{(t)}) - \mathbf{y}^{(t+1)} \right\|^2, \tag{9}$$

where $N_t$ is the number of forecast steps (e.g., $N_t = 3$ for a three-step forecast), $V$ is the set of graph entities (nodes and edges), and $\mathbf{y}^{(t+1)}$ is the ground-truth state at time $t + 1$.

In parallel, we compute the supervised loss, where the model receives ground-truth inputs at each step (Fig. 4a):

$$\mathcal{L}_{\text{real}} = \frac{1}{N_t} \sum_{t=1}^{N_t} \sum_{v \in V} \left\| \hat{\mathbf{y}}^{(t+1)} - \mathbf{y}^{(t+1)} \right\|^2. \tag{10}$$





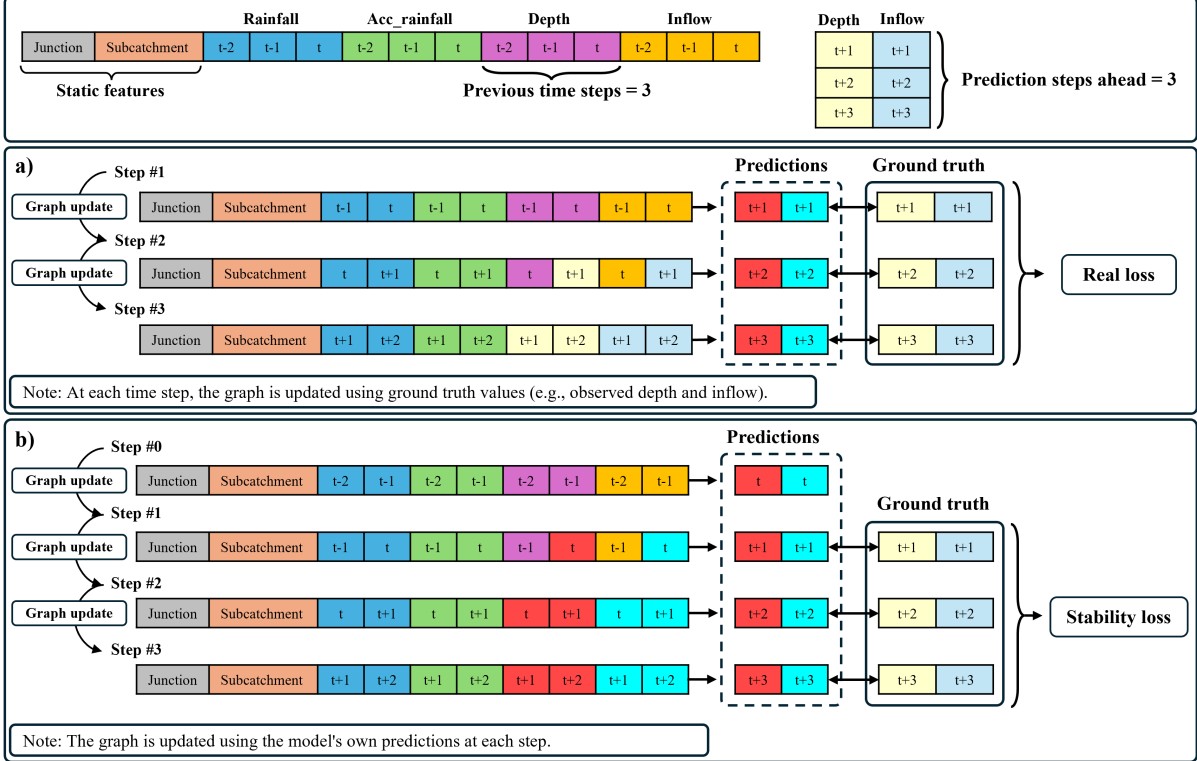

**Figure 4.** Comparison of standard supervised training and pushforward trick for autoregressive stability. **(a)** During standard training, the graph is updated at each step using ground truth observations (e.g., depth and inflow), and the loss is computed between predictions and true values (real loss). **(b)** In pushforward mode, the model updates the graph using its own previous predictions at each step, mimicking inference-time distribution shift. The stability loss is computed based on the discrepancy between rollout predictions and ground truth. This approach improves long-term forecasting robustness by penalizing deviation from true trajectories under autoregressive input conditions.

The combined training loss for autoregressive learning is:

$$\mathcal{L}_{\text{base}} = \mathcal{L}_{\text{real}} + \mathcal{L}_{\text{stability}}. \tag{11}$$

This formulation ensures short-term prediction accuracy while promoting long-term forecasting stability under autoregressive conditions.

## 3.3 Loss function

The overall training objective extends the base loss described in Sec. 3.2 by incorporating two auxiliary loss components that enhance physical interpretability and stability under long rollout horizons. They are defined as follows:





(1) Penalty loss for physical constraints: To prevent physically implausible predictions, this component penalizes negative
water depths at junctions and negative conduit depths along edges during training. These constraint-based penalties ensure that
the model respects basic hydraulic feasibility.

(2) Physics-guided differential consistency loss: This term promotes spatial consistency by penalizing deviations in hydraulic
gradients across adjacent nodes. In stormwater systems, flow behavior is governed by depth and pressure differences between
connected points. To reflect this, the differential loss compares predicted and true differences in water depth and inflow along
conduits:

$$\mathcal{L}_{\text{diff}} = \frac{1}{|E|} \sum_{(i,j) \in E} \left( (\Delta\hat{d}_{ij} - \Delta d_{ij})^2 + (\Delta\hat{q}_{ij} - \Delta q_{ij})^2 \right) \tag{12}$$

Here, $\Delta\hat{d}_{ij} = \hat{d}_i - \hat{d}_j$, $\Delta d_{ij} = d_i - d_j$, and similarly for inflow differences: $\Delta\hat{q}_{ij} = \hat{q}_i - \hat{q}_j$, $\Delta q_{ij} = q_i - q_j$.

Therefore, the total training loss is expressed as:

$$\mathcal{L}_{\text{total}} = \mathcal{L}_{\text{base}} + \mathcal{L}_{\text{penalty}} + \mathcal{L}_{\text{diff}}, \tag{13}$$

During validation, only the base forecasting terms are used to evaluate model accuracy, while auxiliary losses are applied
only during training to guide learning toward physically consistent and stable predictions.

## 3.4  Evaluation Metrics

To assess the performance of the proposed GNN-SWS, several standard evaluation metrics are employed:

- **Root Mean Squared Error (RMSE):**

$$\text{RMSE} = \sqrt{\frac{1}{N_t} \sum_{t=1}^{N_t} (y^t - \hat{y}^t)^2} \tag{14}$$

Measures the average magnitude of prediction errors in the original units.

- **Nash–Sutcliffe Efficiency (NSE):**

$$\text{NSE} = 1 - \frac{\sum_{t=1}^{N_t} (y^t - \hat{y}^t)^2}{\sum_{t=1}^{N_t} (y^t - \overline{y})^2} \tag{15}$$

A commonly used metric in hydrological modeling, where an NSE of 1 indicates perfect prediction.

- **Pearson Correlation Coefficient ($r$):**

$$r = \frac{\sum_{t=1}^{N_t} (y^t - \overline{y})(\hat{y}^t - \overline{\hat{y}})}{\sqrt{\sum_{t=1}^{N_t} (y^t - \overline{y})^2} \sqrt{\sum_{t=1}^{N_t} (\hat{y}^t - \overline{\hat{y}})^2}} \tag{16}$$

Measures the strength and direction of the linear relationship between predictions and observations. $r = 1$ indicates
perfect positive correlation, $r = 0$ indicates no linear correlation, and $r = -1$ indicates perfect negative correlation.





Here, $y^t$ and $\hat{y}^t$ denote the observed and predicted values at time step $t$, respectively; $\overline{y}$ and $\overline{\hat{y}}$ are the mean values of the
observed and predicted time series; and $N_t$ is the total number of time steps used for evaluation.

In addition to continuous predictions, the model's ability to detect flooded nodes is evaluated using binary classification metrics. A node is considered flooded if its predicted water depth exceeds the maximum allowable depth (i.e., the physical depth of the manhole or junction). Flood detection based on predictions are compared against flood status from the SWMM model using the following metrics:

$$\text{Precision} = \frac{\text{TP}}{\text{TP} + \text{FP}} \tag{17}$$

Represents the proportion of correctly predicted flooded nodes out of all nodes predicted as flooded.

$$\text{Recall} = \frac{\text{TP}}{\text{TP} + \text{FN}} \tag{18}$$

Measures the proportion of actual flooded nodes that are correctly identified by the model.

$$\text{Critical Success Index (CSI)} = \frac{\text{TP}}{\text{TP} + \text{FP} + \text{FN}} \tag{19}$$

CSI evaluates the fraction of correctly predicted flooded nodes over the total number of nodes that are either observed or predicted to be flooded.

In all formulas, TP (true positives) refers to nodes correctly predicted as flooded, FP (false positives) are non-flooded nodes incorrectly predicted as flooded, and FN (false negatives) are flooded nodes that the model failed to identify.

## 4   Case study and data preparation

We applied the proposed GNN-SWS framework to the Haven Creek Watershed, a coastal urban drainage system in Norfolk, Virginia, USA (Fig. 5). This watershed experiences frequent pluvial flooding due to its low-lying topography, dense stormwater infrastructure, and tidal influence, making it a representative environment for testing data-driven flood modeling approaches.

The stormwater pipe network was derived from the City of Norfolk's GIS data, with pipe diameters and invert elevations assigned based on municipal design specifications. Being a tidally-influenced watershed, the stormwater pipes are often par-
tially full with seawater due to their low elevation. To account for this and to better capture observed flooding during rainfall events, pipe diameters were reduced to represent diminished flow capacity under dry-weather conditions, where pipes are only partially full.

Crowdsourced flood reports from the Google-owned navigation app Waze during the period 2018 to 2023 were used to identify locations that experience flooding and to inform model the calibration and validation of the SWMM model. Seven
historical storm events were used to calibrate the model against Waze-reported flood locations, with event durations ranging



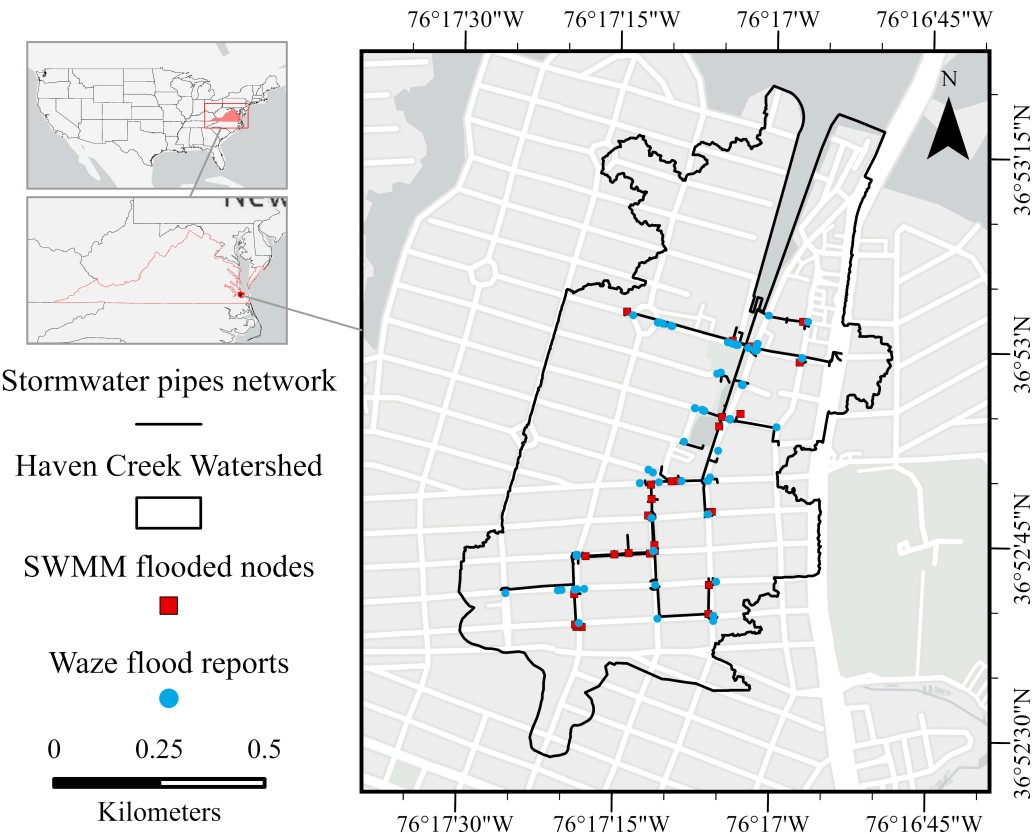

**Figure 5.** Stormwater network and flood observations in the Haven Creek Watershed, Norfolk, VA, USA, including SWMM-predicted flooded nodes (red) and Waze reports (blue).

from 6 to 26 hours and total rainfall between 22.1 and 153.7 mm, corresponding to return periods from 1 to 8.8 years. Sensitivity analyses were performed on key subcatchment parameters—width ($\pm75\%$), slope ($\pm10\%$), infiltration rate ($\pm25\%$), and imperviousness ($\pm25\%$)—to assess model robustness. Across all scenarios, the model successfully reproduced flooding in locations reported by Waze.

A total of 85 storm events from 2018 to 2023 were collected from the Norfolk International Airport station (approximately 7 km from the watershed). Events were selected based on a minimum duration of 6 hours, hourly rainfall continuity, and total rainfall exceeding 15 mm. To expand the range of rainfall intensities and support generalization within the GNN framework, each event was scaled by factors of 1.2, 1.4, or 1.6. This augmentation process resulted in a final dataset of 300 storm events, which was used for training, validation, and testing of the proposed GNN-SWS model.

Out of the 300 storm events, 36 were held out as the test set for final evaluation. The remaining 264 events were used to generate input–output graph samples using a sliding window approach (see Sec. 3.1). These graph samples were then pooled and randomly split into training (85%) and validation (15%) sets.





## 5    Results and discussion

This section presents a comprehensive analysis of the model's performance across various aspects of the problem. It begins
with the configuration of the GNN-SWS model, including the selection and tuning of key hyperparameters. The model's
predictive capability is then assessed through hydraulic state forecasting at the node level (inflow and depth), followed by
performance evaluation at the edge level (flow and depth in conduits), and an overall statistical analysis across all test events.
Beyond pointwise predictions, the model's ability to detect flooded nodes during intense rainfall events is examined. To gain
deeper insights into spatial uncertainty, we analyze how prediction errors are distributed across the network over time. Finally,
we discuss the limitations of the current GNN-SWS framework and outline potential directions for future enhancement and
broader applicability.

### 5.1    Model configuration

The GNN-SWS model architecture and training configuration are summarized in Table 1. The encoder maps raw node and
edge inputs into latent representations of dimension 64 using MLPs with three layers and a hidden size of 64. The processor
consists of five GNN layers, each equipped with residual connections to facilitate gradient flow and improve convergence
during training. Each GNN layer incorporates an internal MLP with a hidden size of 64 and three layers to update node and
edge representations. The decoder maps the final latent embeddings to the target variables using a 3-layer MLP with the same
hidden size.

Training is performed using a batch size of 128 and the Adam optimizer (Kingma, 2014), with an initial learning rate of
0.001. A learning rate scheduler was employed to decay the step size by a factor of 0.999 every 64 batches, promoting smoother
convergence during training. Hyperparameters were selected based on manual tuning and preliminary experiments to ensure
stable training dynamics and strong predictive performance across multiple test events. The model was trained on a workstation
equipped with an NVIDIA RTX 4090 GPU and 128 GB of RAM.

### 5.2    Hydraulic State Prediction Performance

To evaluate the spatial performance of the GNN-SWS model in predicting hydraulic states across the stormwater network,
we computed the NSE at each node and edge for a representative storm event with a duration of 19 hours and an average
rainfall intensity of approximately 9 mm/hr. This event was selected as a severe rainfall scenario, characterized by sustained
precipitation and multiple high-intensity bursts, which resulted in widespread activation of the stormwater system.

As shown in Figures 6 and 7, the model achieves consistently high performance across most of the network, with the majority
of nodes and conduits demonstrating excellent agreement between predicted and observed values. These results confirm that
the GNN-SWS surrogate is capable of accurately reproducing both nodal (depth and inflow) and edge-level (flow and depth)
dynamics.

Localized reductions in model performance are observed at a subset of outer nodes, particularly those situated near the
network boundary or with limited hydraulic connectivity. From a graph learning perspective, these nodes have restricted access





**Table 1.** Architecture and training configuration for the proposed GNN-SWS model.

| Component | Name | Description |
|-----------|------|-------------|
| Encoder | Latent dimension | Size of initial node and edge embeddings: **64** |
| | MLP layers | Number of layers in each MLP for node and edge embeddings: **3** |
| | MLP hidden size | Hidden size used in each MLP layer: **64** |
| Processor | GNN layers | Number of message-passing steps: **5** |
| | Residual connections | Whether residual connections are used in GNN layers: **True** |
| | MLP (hidden size / layers) | Internal MLPs within each GNN block use hidden size = **64**, layers = **3** |
| Decoder | MLP structure | MLP with **3** layers and hidden size = **64** |
| Training | Batch size | Number of training windows used in each batch: **128** |
| | Learning rate | Step size for gradient descent optimization: **0.001** |
| | Scheduler | Learning rate decays by factor **0.999** every **64** batches |

to neighboring information during the message-passing process, resulting in less informative latent representations and lower prediction accuracy. This effect is a known limitation in GNN architectures, where node performance is influenced by the richness and depth of the message-passing neighborhood.

Additionally, a comparison between Figures 6a and 6b reveals that low-performing nodes are more prevalent in the inflow predictions than in depth. This discrepancy is likely due to the extremely small inflow values observed at peripheral locations (e.g., inflows $< 0.005$ m$^3$/s), where minor absolute errors can result in disproportionately low NSE values due to the metric's sensitivity to relative deviations. From a stormwater management perspective, these low-flow nodes are typically less critical for system-level performance and may have limited practical importance compared to mainline or outlet locations.

The edge-level predictions (Fig. 7) exhibit highly uniform performance, with nearly all conduits achieving NSE values indicative of strong model agreement. This reflects the model's ability to accurately capture flow routing and hydraulic gradients along conduit links, even in spatially extended sections of the network.

Overall, the GNN-SWS model demonstrates highly promising spatial generalization and fidelity in replicating hydraulic states across a complex urban stormwater system.

### 5.2.1 Node-Level Performance: Inflow and Depth

To assess the temporal performance of the GNN-SWS model at the node level, Figures 8 and 9 present time series predictions for water depth and inflow, respectively, at selected junctions under three representative storm events. Each column corresponds to a distinct event: **(a)** Event-I (moderate duration and intensity; 12.8 hr, 2.30 mm/hr), **(b)** Event-II (short duration, high intensity; 7 hr, 7.17 mm/hr), and **(c)** Event-III (long duration, moderate intensity; 38 hr, 2.31 mm/hr). Rainfall inputs are shown in the top row of each column to provide context for the system's hydraulic response. This range enables evaluation of the model's robustness under varying hydrological conditions.



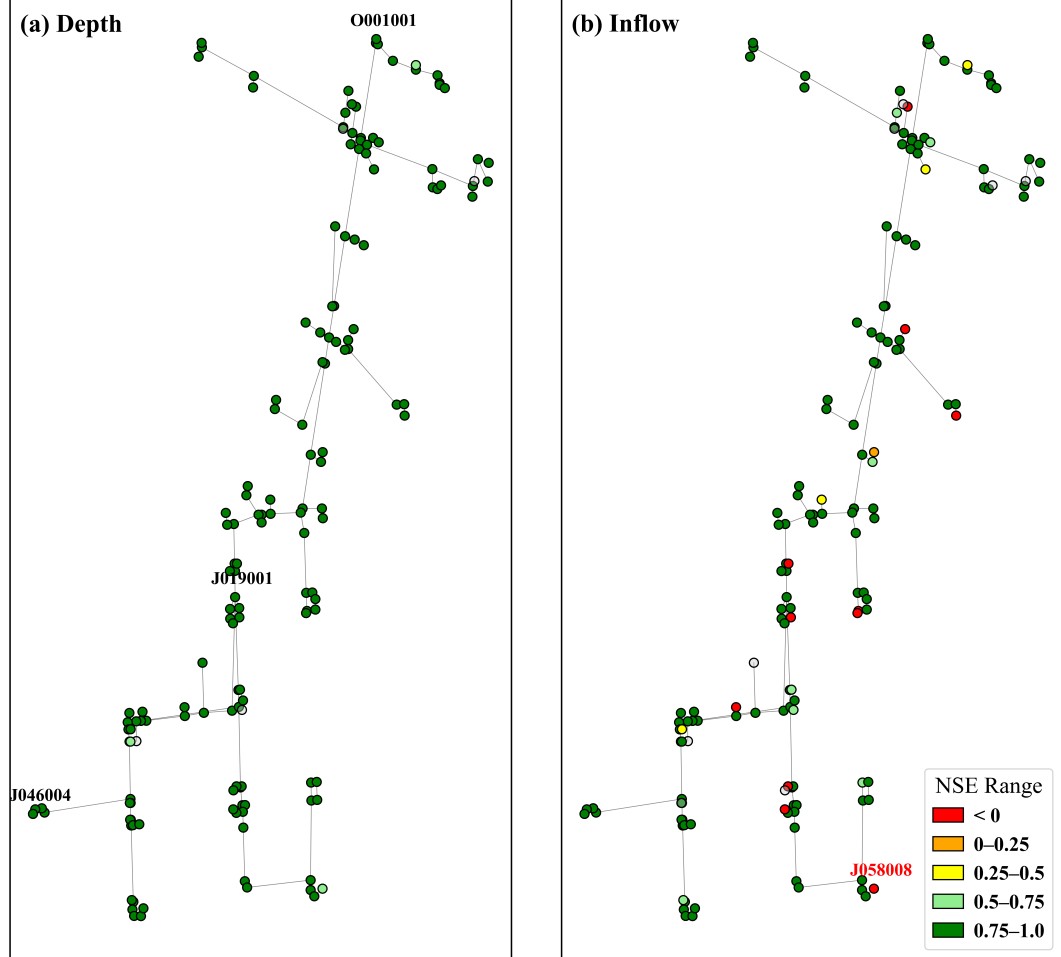

**Figure 6.** Node-level performance of the GNN-SWS model based on Nash–Sutcliffe Efficiency (NSE) scores. **(a)** NSE for depth prediction at junctions. **(b)** NSE for inflow prediction at junctions. Higher values (green) indicate better model performance, while lower values (red) highlight areas where predictions are less accurate.

Each row represents a different spatial location within the drainage network, progressing from upstream (J046004) to midstream (J019001) to downstream (outfall-O001001) junctions. The spatial location of these nodes is illustrated in Fig. 6a.

Notably, the model's predictive accuracy improves from upstream to downstream nodes. This trend is driven by the increasing flow accumulation and smoother hydraulic responses observed in downstream sections of the network. These nodes tend to exhibit higher average inflow and depth, which enhances signal-to-noise ratio and facilitates more stable learning during model training. For example, downstream junctions in Figures 8 and 9 show consistently higher magnitude responses and smaller relative deviations. In contrast, upstream nodes—particularly those with small contributing areas—tend to exhibit lower inflow and depth magnitudes (e.g., mean inflow < 0.005 m$^3$/s), which are inherently more difficult for the model to learn. These





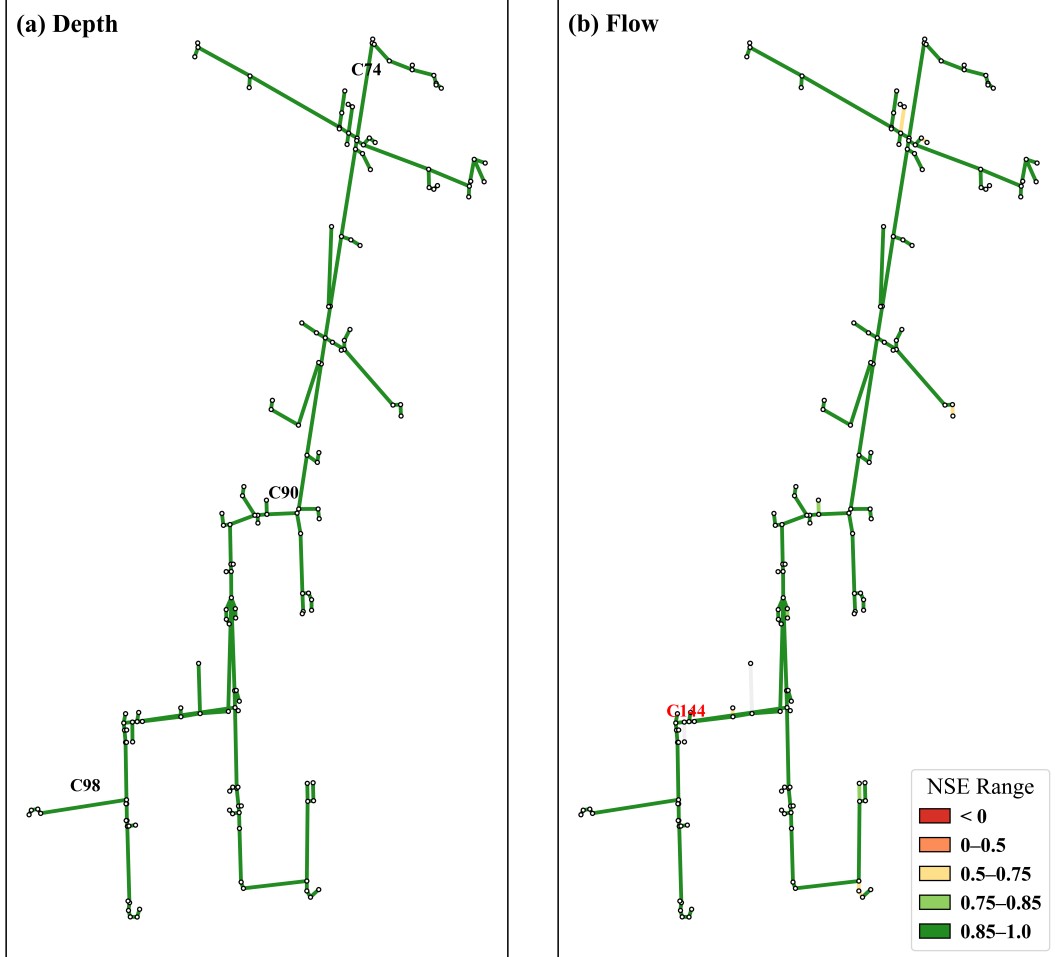

**Figure 7.** Edge-level performance of the GNN-SWS model based on Nash–Sutcliffe Efficiency (NSE) scores. **(a)** NSE for depth prediction in conduits. **(b)** NSE for flow prediction in conduits. Green tones indicate high predictive accuracy (NSE closer to 1), while warmer colors mark sections with relatively lower performance.

low-magnitude responses provide weak hydraulic signals, often lacking sharp temporal gradients or spatial propagation cues, making it challenging for the GNN (or any other surrogate model) to extract meaningful patterns during training.

Nevertheless, the GNN-SWS model consistently demonstrates strong agreement with the SWMM simulations across all events and node locations. This consistency reflects the effectiveness of the model architecture and learning strategy, which is designed to capture the spatiotemporal dynamics of stormwater systems. The model is trained to predict both water depth and inflow at junctions (and flow and depth at the conduit level) for three future time steps (i.e., $t+1, t+2, t+3$), using information from the previous three time steps (i.e., $t-2, t-1, t$) as input. These spatiotemporal graphs are constructed using a sliding-

window approach, resulting in a sequence of training samples per event that reflect evolving system dynamics. Each graph





**Figure 8.** Time series comparison of water depth predictions by the GNN-SWS model (dashed red lines) and SWMM simulations (solid black lines) at selected junctions under three representative storm events. Each column corresponds to a distinct event: **(a)** Event-I: moderate-duration (12.8 hr, 2.30 mm/hr), **(b)** Event-II: intense short-duration (7 hr, 7.17 mm/hr), and **(c)** Event-III: long-duration moderate-intensity (38 hr, 2.31 mm/hr). Each row represents a different junction location along the drainage network: upstream (J046004), midstream (J019001), and downstream (outfall-O001001). Rainfall input is shown in the top row for each event. Performance metrics (Mean depth, RMSE, NSE, and r) are reported for each node to evaluate temporal accuracy.

aggregates both dynamic inputs—such as rainfall, accumulated rainfall, depth, inflow, and flow—and static attributes of nodes, conduits, and subcatchments, including elevation, slope, imperviousness, and geometry.





During inference, the model is provided only with the dynamic inputs from the initial three time steps and then generates each future prediction recursively, using its own output from one step as input for the next. This autoregressive rollout strategy
reflects a realistic forecasting scenario where future ground-truth values are unavailable. The only external information updated at each time step is the rainfall input, which acts as the primary boundary condition driving the hydraulic response.

The GNN-SWS model accurately captures the temporal evolution of both inflow and depth. It successfully reproduces the timing of onset, magnitude of peak flows, and the recession limbs of hydrographs across a variety of hydrological conditions. This strong performance is consistently supported by high values of the NSE, low RMSE, and near-perfect r, demonstrating
the model's capacity to learn and generalize spatiotemporal hydraulic behavior across different locations and event types.

### 5.2.2 Edge-Level Performance: Flow and Depth

Figures 11 and 10 present the predicted and simulated flow and depth time series across the three representative storm events. As before, conduits are selected to represent upstream (C98), midstream (C90), and downstream (C74) locations, allowing for spatially distributed evaluation of the model's routing performance.
The GNN-SWS model demonstrates strong agreement with the physics-based SWMM solver in both flow magnitude and temporal dynamics. Key hydraulic behaviors, including peak timing, wave translation, and attenuation, are accurately captured across the full length of the network. Downstream segments, which exhibit smoother hydrographs due to cumulative inflows and extended travel times, show particularly high model fidelity, with consistently high NSE values. This reinforces the model's capacity to learn and generalize the integrated effects of upstream runoff and conduit hydraulics.
In upstream conduits, where flow responses are more intermittent and low in magnitude, the model retains good predictive skill but occasionally exhibits greater relative error. These segments are more sensitive to small perturbations in upstream inputs and local network structure. Nevertheless, even in these low-flow regimes (e.g., mean flow $< 0.005$ m$^3$/s), the GNN-SWS model maintains accurate timing and qualitative trends, which are critical for informing early warning systems and distributed control strategies.
Importantly, the model's ability to predict flow and depth jointly along edges demonstrates its effectiveness at capturing the dynamic mass balance and energy dissipation mechanisms that govern urban hydrodynamics. This is especially valuable in operational contexts, where real-time flow predictions at key conduits are used for flood routing, system optimization, or adaptive control of low-impact development (LID) measures.

### 5.2.3 Overall Performance Analysis

While the previous sections demonstrate the model's high fidelity for representative storm events, it is crucial to evaluate its performance statistically across the entire test set to confirm its robustness. To this end, the model's predictions for all 36 test events are compared against the SWMM simulations. The aggregated results are presented in Fig. 12 as box plots, summarizing the distribution of NSE and Normalized RMSE metrics for the four primary output variables.

The NSE results (Fig. 12a) demonstrate exceptionally strong performance, particularly for Junction-Depth, Conduit-Depth,
and Conduit-Flow, with median NSE values around 0.99 and narrow interquartile ranges. This indicates highly consistent



**Figure 9.** Time series comparison of inflow predictions by the GNN-SWS model (dashed red lines) and SWMM simulations (solid black lines) at selected junctions under three representative storm events. Each column corresponds to a distinct event: **(a)** Event-I: moderate-duration (12.8 hr, 2.30 mm/hr), **(b)** Event-II: intense short-duration (7 hr, 7.17 mm/hr), and **(c)** Event-III: long-duration moderate-intensity (38 hr, 2.31 mm/hr). Each row represents a different junction location along the drainage network: upstream (J046004), midstream (J019001), and downstream (outfall-O001001). Rainfall input is shown in the top row for each event. Performance metrics (Mean inflow, RMSE, NSE, and r) are reported for each node to evaluate temporal accuracy.

predictions and minimal deviations across nearly all test events. Junction-Inflow predictions exhibited a slightly lower median







**Figure 10.** Time series comparison of conduit depth predictions by the GNN-SWS model (dashed red lines) and SWMM simulations (solid black lines) for selected conduits under three representative storm events. Each column corresponds to a distinct event: **(a)** Event-I: moderate-duration (12.8 hr, 2.30 mm/hr), **(b)** Event-II: intense short-duration (7 hr, 7.17 mm/hr), and **(c)** Event-III: long-duration moderate-intensity (38 hr, 2.31 mm/hr). Each row represents a different conduit location along the drainage network: upstream (C98), midstream (C90), and downstream (C74). Rainfall input is shown in the top row for each event to provide context for the system's hydraulic response. Performance metrics (Mean depth, RMSE, NSE, and $r$) are reported for each conduit to evaluate temporal prediction accuracy.

NSE ($\sim$0.98) and greater variability, reflecting challenges in accurately modeling very small inflow magnitudes at peripheral junctions.





**Figure 11.** Time series comparison of conduit flow predictions by the GNN-SWS model (dashed red lines) and SWMM simulations (solid black lines) for selected conduits under three representative storm events. Each column corresponds to a distinct event: **(a)** Event-I: moderate-duration (12.8 hr, 2.30 mm/hr), **(b)** Event-II: intense short-duration (7 hr, 7.17 mm/hr), and **(c)** Event-III: long-duration moderate-intensity (38 hr, 2.31 mm/hr). Each row represents a different conduit location along the drainage network: upstream (C98), midstream (C90), and downstream (C74). Rainfall input is shown in the top row for each event to provide context for the system's response. Performance metrics (Mean flow, RMSE, NSE, and $r$) are reported for each conduit to evaluate temporal prediction accuracy and routing behavior.

Similarly, NRMSE distributions (Fig. 12b) confirm these observations. Depth-related variables consistently achieved very low median errors (NRMSE < 0.1), further highlighting the accuracy of depth predictions. Junction-Inflow and Conduit-Flow






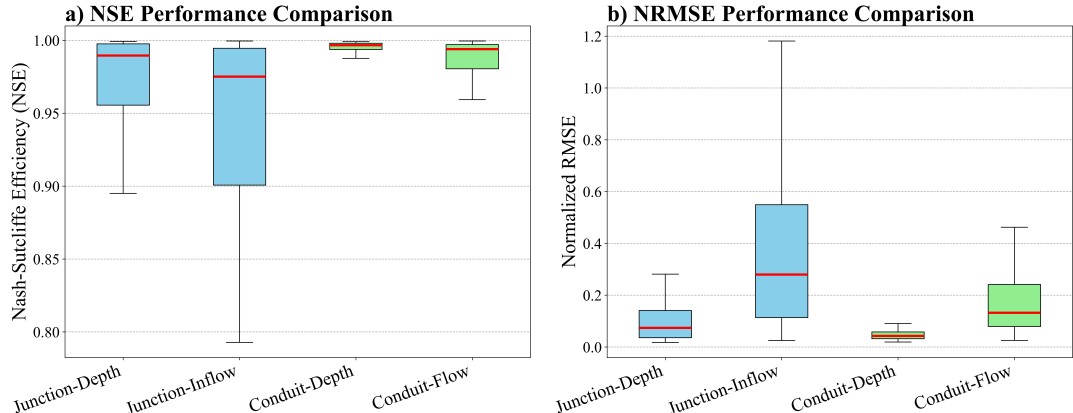

**Figure 12.** Overall performance of the GNN-SWS model across all test events. Boxplots show the distribution of (a) Nash-Sutcliffe Efficiency (NSE) and (b) Normalized Root Mean Squared Error (NRMSE) for the four key hydraulic variables. The red line in each box indicates the median value.

displayed broader NRMSE distributions with higher median values due to the inherently low magnitude of inflow and flow in certain parts of the network. These small-magnitude conditions magnify relative prediction errors, thereby causing wider variability in normalized metrics.

Overall, this statistical evaluation confirms that the GNN-SWS model maintains excellent accuracy and robustness across
diverse storm conditions, effectively capturing both node-level and conduit-level hydraulic states in urban stormwater systems.

### 5.3   Flooded Node Detection Performance

Beyond continuous state prediction, the GNN-SWS model was evaluated for its ability to detect flooding conditions at the node level, which is critical for issuing alerts, activating control responses, and prioritizing maintenance across urban drainage networks.

Figure 13 presents the spatial distribution of predicted flooded nodes across the network for a representative storm event. This visualization enables qualitative interpretation of the model's ability to capture localized flood conditions under realistic hydrological forcing.

Flood status at each node was inferred by comparing the predicted depth values to the node-specific maximum depth thresholds defined in the network configuration. To mitigate the effect of numerical fluctuations around the threshold, a two-band
classification logic was adopted using a tolerance margin of $\varepsilon = 0.04$ m. If a node was flagged as flooded at any time during the simulation, it was labeled as flooded for the entire event. According to SWMM, flooding refers to all water that overflows a node, whether it ponds or not, and includes surcharge and overflow conditions. Based on this reference, we computed the CSI, precision, and recall. For a representative high-intensity, long-duration storm event—with an average intensity of 9.07 mm/hr, and a total duration of 19.0 hours—the model achieved a CSI of 0.77, precision of 0.90, and recall of 0.84. These



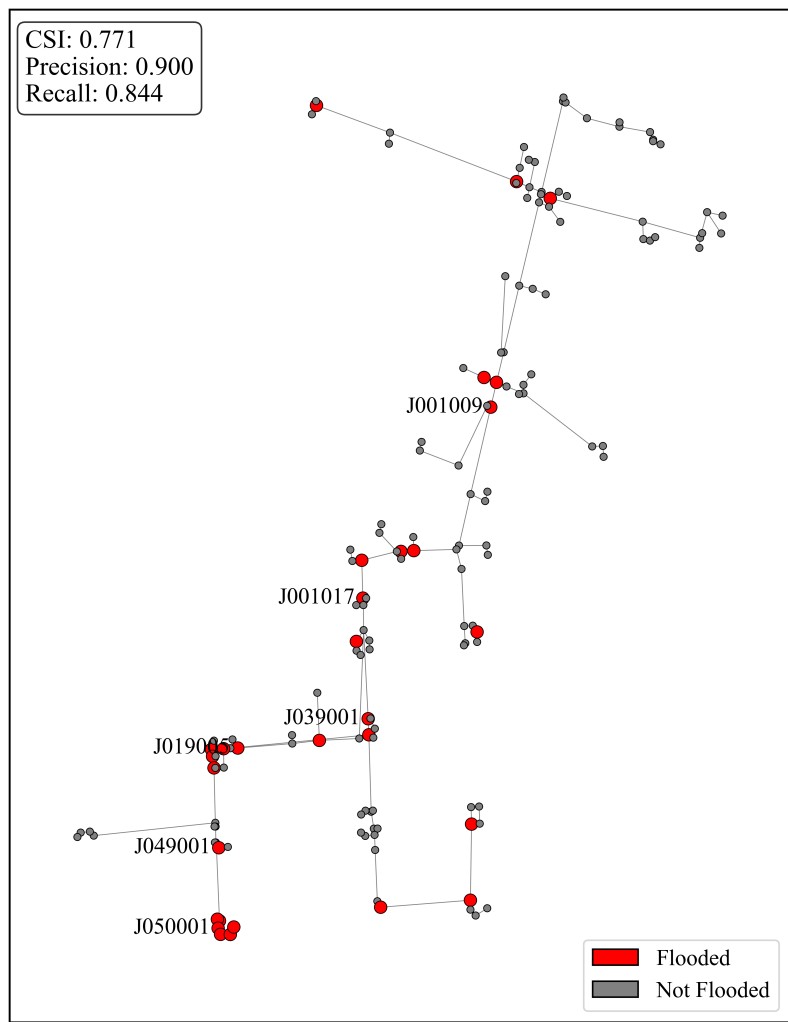

**Figure 13.** Spatial distribution of flooded nodes predicted by the GNN-SWS model for a representative storm event. Nodes identified as flooded are shown in red, while non-flooded nodes are displayed in grey.

values indicate that the GNN-SWS model can accurately classify flooded junctions with both high specificity and sensitivity, making it a suitable surrogate for operational flood detection.

In addition to binary classification, we evaluated the GNN-SWS model's ability to approximate flood volume at key junctions and compared it with node-level flood volumes reported by SWMM. Flood volume at each node was estimated by computing the cumulative difference between predicted inflow and total outflow via connected downstream conduits, aggregated over time

steps in which the node was classified as flooded.

To ensure a focus on hydraulically meaningful conditions, we restricted this comparison to nodes with SWMM-reported flood volumes exceeding $0.05 \times 10^6$ liters. As shown in Table 2, the GNN-SWS model closely matched the SWMM-predicted



**Table 2.** Comparison of flood volume (in $10^6$ liters) between the physics-based SWMM model and the surrogate GNN-SWS model at major flood locations (volume $> 0.05$ $10^6$ L).

| Node | SWMM Volume ($10^6$ L) | GNN-SWS Volume ($10^6$ L) | % Error |
|---------|------------------------|---------------------------|---------|
| J001009 | 1.943 | 2.023 | 4.14 |
| J001017 | 0.777 | 0.819 | 5.46 |
| J019005 | 0.615 | 0.616 | 0.15 |
| J039001 | 0.468 | 0.504 | 7.60 |
| J049001 | 0.166 | 0.163 | -2.05 |
| J050001 | 0.063 | 0.108 | 71.53 |

flood volumes at major flood hotspots. For example, at nodes J001009 and J019005, the relative error in flood volume was only 4.1% and 0.2%, respectively. These nodes are typically situated near flow convergence zones or downstream reaches where large volumes tend to accumulate. While larger relative errors were observed at some locations (e.g., 71.5% at J050001), such cases often correspond to nodes located in upstream or peripheral areas of the network where the absolute flood volumes are low. In these scenarios, even small discrepancies in predicted values can produce large relative errors, which are less critical from an operational standpoint.

### 5.3.1  Temporal Error Distribution

To evaluate model performance over time, we calculate the temporal distribution of absolute errors across the entire network. At each time step, the median absolute error is computed over all nodes (or all edges), along with the 25th and 75th percentiles to characterize variability. This approach is applied separately for depth and inflow at nodes, and for depth and flow at conduits. The resulting distributions are presented in Fig. 14, where the first row shows the rainfall distribution for each event, the second row corresponds to node-level variables, and the third row presents edge-level variables. Two storm events are analyzed in this comparison: event-I, a moderate rainfall event with a duration of 12.83 hours and an average intensity of 6.92 mm/hr; and event-II, a more extreme scenario with a duration of 25.0 hours and a higher average intensity of 12.03 mm/hr.

The temporal error profiles in Fig. 14 highlight the robust performance of the model across both node- and edge-level variables. For the full simulation period, median absolute errors remain consistently low, with narrow interquartile ranges (25th–75th percentiles), indicating stable and accurate predictions throughout the network. This stability is maintained across both storm events, despite their differing rainfall magnitudes and durations.

Across both events, node-level variables—depth (Fig. 14b) and inflow (Fig. 14c)—and edge-level variables—depth (Fig. 14d) and flow (Fig. 14e)—exhibit comparable error magnitudes and temporal patterns. While slight variability exists across variables and between storm events, all predictions remain within acceptable error margins. Notably, error magnitudes tend to increase during and shortly after peak rainfall, particularly for the more intense Event-II, reflecting the system's sensitivity to dynamic





**Figure 14.** Temporal distribution of absolute errors across the network for two storm events: Event-I (blue) and Event-II (red). The top row (a) shows the rainfall distribution over time for both events. The second row presents absolute errors for depth (b) and inflow (c) at nodes, while the third row shows errors for depth (d) and flow (e) at conduits. Median errors with interquartile ranges (25th–75th percentiles) illustrate the model's predictive stability over time.

loading. Despite this, the model maintains predictive reliability even during peak loading, and its performance remains stable throughout the extended duration of the longer storm event.





### 5.3.2 Model Limitations and Practical Constraints

Despite the overall strong performance of the GNN-SWS model, some limitations emerge in boundary regions of the drainage network, as discussed in Sec. 5.2. These include nodes and conduits located near the network periphery, which tend to exhibit lower predictive accuracy compared to centrally located components. As illustrated in Fig. 15, both the selected boundary node (see Fig. 6b) and conduit (see Fig. 7b) exhibit rather large discrepancies between predicted and observed values, particularly for variables with low magnitudes such as inflow and flow.

This limitation reflects a common challenge in spatial modeling where boundary components often lack sufficient neighboring context, regardless of the modeling approach. In GNNs, this manifests as reduced information exchange during message passing due to fewer adjacent nodes. As a result, the learned embeddings for these nodes and edges may carry less contextual information, which can contribute to slightly lower prediction accuracy in peripheral areas of the network.

From a hydrodynamic standpoint, boundary nodes and conduits are often associated with low flow rates and shallow depths, particularly during dry conditions or under low-intensity rainfall. In such settings, even small absolute errors can yield relatively large deviations in normalized error metrics. For example, as shown in Fig. 15, the inflow and flow predictions have negative NSE values despite low RMSE scores, primarily due to the small mean values of the observed targets. These localized inaccuracies, however, have limited impact on the overall predictive reliability of the model, which remains high across the main structure of the network.

Regarding the GNN-SWS structure, it operates as an end-to-end surrogate that predicts node- and conduit-level hydraulic states directly from rainfall input. However, the model's predictions are inherently constrained by the temporal resolution of the input data. For instance, when trained on 10-minute rainfall intervals, the model produces hydraulic outputs at the same frequency.

This fixed-resolution behavior limits the model's flexibility in real-world applications, where rainfall data may be available at coarser or irregular intervals, but high-resolution hydraulic outputs (e.g., every 5 or 10 minutes) are still required. Addressing this constraint would necessitate retraining on appropriately resampled data or integrating temporal interpolation or multi-resolution capabilities within the network architecture.

## 6 Interactive Dashboard for Rapid Hydraulic State Exploration and System Analysis

To enhance the usability and practical value of the proposed GNN-SWS surrogate model, we developed an interactive dashboard that enables real-time exploration and visualization of predicted hydraulic states across a real urban stormwater network.

The dashboard serves as an efficient post-processing tool, allowing users to analyze stormwater system dynamics without the need to rerun the full physics-based SWMM simulations.

Key functionalities include:

- **Event switching:** Explore different storm events to observe how the network responds to various rainfall intensities and durations.



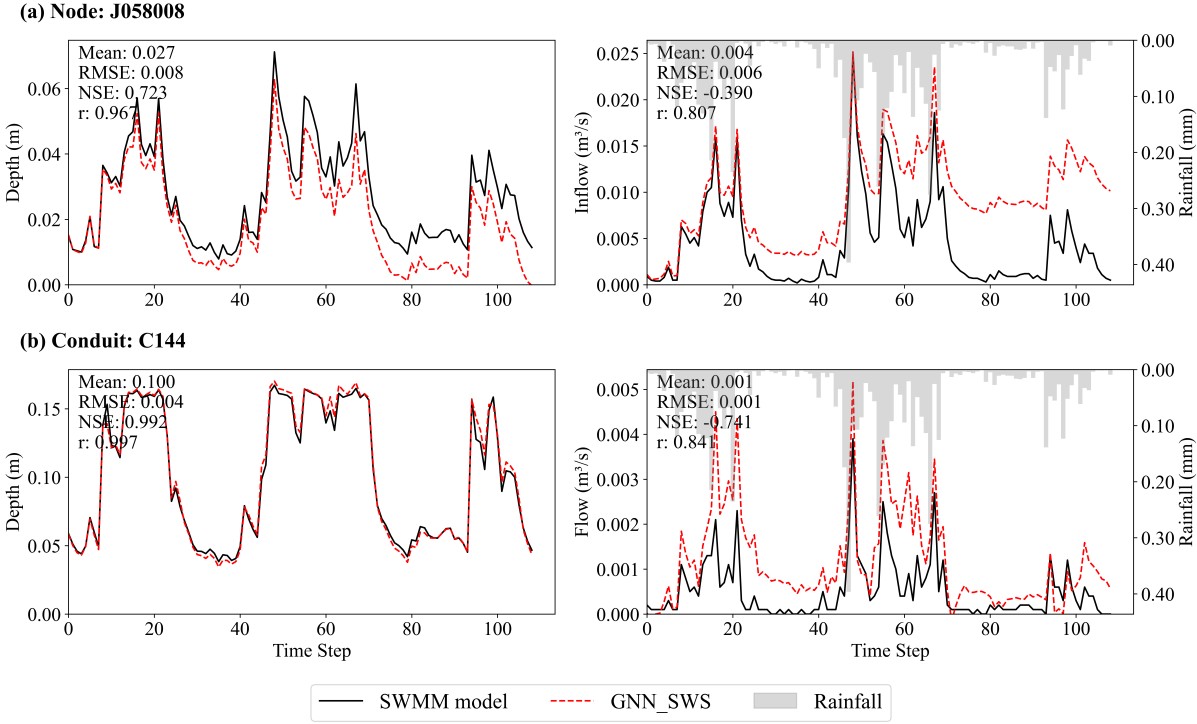

**Figure 15.** Time series of predicted and observed hydraulic variables at a representative boundary node and conduit, illustrating localized prediction errors in peripheral regions of the network.

- **Domain selection:** Toggle between junctions (nodes) and conduits (edges) to evaluate different parts of the system.

- **Time series visualization:** Inspect temporal evolution of predicted hydraulic variables (e.g., depth, inflow, flow) for selected nodes or edges.

- **Evaluation metrics:** View RMSE, NRMSE, Rel_MAE, $r$, and NSE values computed both for individual junctions or conduits over the full simulation period, and for the entire network at each time step.

This dashboard is tailored to the stormwater network of the Haven Creek Watershed in Norfolk, VA, and is built on a
500 curated set of historical storm events. It is primarily intended as an educational resource to support learning in stormwater-related courses and research. It enables users to explore system dynamics, track flow behavior during critical periods, identify vulnerable nodes, and evaluate surrogate model performance. Users can visually compare GNN-SWS surrogate predictions against physics-based SWMM simulations, assess system responses under varying storm conditions, and examine error metrics across time and space. This interactive environment provides a valuable platform for both instructional and exploratory scenario
analysis.





## 7 Conclusions

Urban pluvial flooding poses a significant challenge exacerbated by climate variability and rapid urbanization, necessitating timely and accurate hydraulic predictions that traditional physics-based models often struggle to provide due to computational demands. To address this, this study developed GNN-SWS, a novel end-to-end graph neural network surrogate model designed 510 for rainfall-driven, real-time hydraulic prediction in stormwater systems.

The GNN-SWS model uniquely learns rainfall-driven inflow dynamics and flow routing directly from rainfall inputs, addressing a critical limitation of previous research (Garzón et al., 2024; Zhang et al., 2024) that relied on precomputed runoff inputs. This enables the prediction of hydraulic states at both junction and conduit levels, overcoming the common focus on nodal outputs alone. Furthermore, a heterogeneous message-passing architecture distinguishes node types for structured hy-515 drologic representation, and an autoregressive forecasting structure supports multi-step predictions over extended horizons. Its architecture also incorporates physics-guided constraints and a pushforward training strategy to ensure physical consistency and enhance long-term forecast stability.

The GNN-SWS model exhibits robust performance in experimental evaluations, demonstrating high accuracy in predicting water depth and inflow at nodes, as well as flow and depth in conduit across diverse storm events. It also proves highly 520 effective in identifying flooded nodes with strong classification and volumetric approximation capabilities at critical locations. Furthermore, temporal analysis underscores the model's consistent predictive stability, even during peak rainfall conditions.

Despite its strong performance, the model exhibits localized reductions in accuracy, particularly at certain boundary nodes and peripheral conduits. This can be attributed to these components having fewer adjacent connections, which limits the extent of information exchange during the message-passing process and can result in less informative latent representations. This 525 is consistent with the overall performance analysis, which confirms that such limitations are localized and do not undermine the model's overall predictive reliability. Additionally, the model's fixed temporal resolution, tied to the input data, restricts its flexibility when rainfall data are available at irregular or coarser intervals. Future work will focus on addressing these limitations by exploring the integration of temporal interpolation or multi-resolution capabilities within the network architecture to enhance flexibility in handling varied input data resolutions.

Additionally, future research direction can include leveraging the proposed method for identifying optimal flood management strategies, specifically to assess the implementation of Low Impact Developments (LIDs) for flood mitigation. The underlying predictive power of the GNN-SWS model makes it a strong foundation for such applications, as it can be adapted to consider hydrological processes in greater detail, accounting for LIDs that alter impervious areas, and ultimately supporting intelligent and adaptive planning of urban stormwater systems for flood resilience.

*Code and data availability.* The source code, data, and related files can be found in the repository:
   https://github.com/Zanko-WRM/GNN-Surrogate-for-Stormwater-Systems.



*Code and data availability.* The source code, data, and related files are available at: https://github.com/Zanko-WRM/GNN-Surrogate-for-Stormwater-Systems. This repository was created and maintained by Zanko Zandsalimi. For inquiries, contact: `mfx2uq@virginia.edu`.

*Author contributions.* **Zanko Zandsalimi:** Conceptualization, Formal analysis, Data curation, Investigation, Methodology, Validation, Visu-
alization, Writing – original draft. **Mehdi Taghizadeh:** Data curation, Investigation, Methodology, Writing – review  editing. **Savannah Lee Lynn:** Data curation, Software, Methodology, Writing – review  editing. **Jonathan L. Goodall:** Supervision, Validation, Writing – review editing. **Majid Shafiee-Jood:** Supervision, Validation, Writing – review  editing. **Negin Alemazkoor:** Conceptualization, Methodology, Supervision, Funding acquisition, Validation, Writing – review  editing.

*Competing interests.* The authors declare that they have no known competing financial interests or personal relationships that could have
appeared to influence the work reported in this paper.



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
