# Peer review of "End-to-End Graph Neural Networks for Real-Time Hydraulic Prediction in Stormwater Systems"

_EGUsphere, 2025_

## Author Comment (AC2)

The paper presents a graph neural network model to surrogate the node and edge hydraulic variables in a sewer system.

On a small case study in Virginia, USA, the model shows almost perfect performance across most predicted metrics.

Despite the paper being in a decent state, I have several major concerns regarding the novelty of the paper.

We thank the reviewer for the time and care invested in this review. Your comments have been valuable in prompting us to clarify the presentation of our contributions, to strengthen the empirical support for some components of the model, and to prepare concrete revisions that will improve the manuscript.

That said, we must respectfully disagree with the conclusion that our work lacks novelty. We recognize, however, that aspects of our manuscript did not clearly articulate the scope and implications of our contributions, which may have contributed to this perception. Across the five major concerns, the review frames our contributions as incremental variations of existing approaches. However, as our detailed responses demonstrate, the novelty is not technical tinkering but a re-formulation of the problem itself: developing a rainfall-driven, end-to-end surrogate for stormwater systems that enables applications outside the reach of prior runoff-driven or hydraulicsonly models. First, unlike runoff-driven surrogates, our rainfall-driven design unifies hydrologic and hydraulic processes, eliminating reliance on external models and enabling applications that partial surrogates cannot support. Second, our heterogeneous graph formulation is a structural innovation that mirrors the physical distinction between subcatchment nodes and routing nodes, preserving hydrologic integrity and enabling localized interventions such as LID optimization. Third, our physics-guided learning strategy, including the stability loss and pushforward formulation, provides a principled way to balance accuracy and robustness during autoregressive rollout, going beyond existing multi-step losses by explicitly detaching gradients to enforce error recovery.

Taken together, along with the other clarifications and revisions outlined in this response, these elements demonstrate that our framework is not a minor variation of existing models but a distinct step forward. Its novelty lies as much in scope and application as in architecture, creating new system-level capabilities, such as integrated flood modeling and infrastructure optimization, that cannot be achieved with partial surrogates.

With this context, we address each of your comments in detail below. Our responses clarify where our approach differs substantively from prior work, explain the rationale for our design choices, and outline the specific revisions we will make to improve clarity and rigor.

The authors claim that the paper has 5 main novelties, but they are either not novel or the results are not validated enough, as I justify hereafter.

1) The model converts rainfall to hydraulic variables at junctions and conduits. There already exists a paper that models both node and edge variables (Garzon et al., 2024b), despite your paper predicts different variables. The main difference consists in directly taking rainfall as an input rather than the runoff generated by a hydrological model. However, this introduces other issues that are not discussed in the paper, such as including all of the hydrological characteristics of each catchment in the proper training of the model.

The reviewer characterizes our contribution as "taking rainfall as input," which downplays the substantive distinction between a runoff-driven surrogate and a rainfall-driven surrogate. Prior GNN-based studies emulate only the hydraulic routing given pre-computed runoff; they therefore remain dependent on a separate hydrologic model and do not remove a major computational and operational bottleneck. Our model, in contrast, is end-to-end. That is, it emulates the complete rainfall-to-hydraulics transformation within a single framework. These are fundamentally different problem formulations, and reducing our contribution to a minor input change misrepresents the novelty of our proposed end-to-end approach.

This end-to-end design is important because it enables capabilities unavailable to runoff-driven surrogates: (i) direct coupling with forecast rainfall for real-time operations without invoking a hydrologic pre-processor; (ii) explicit sensitivity to subcatchment attributes (imperviousness, slope, storage, connectivity), which allows evaluation of land-use and LID interventions; and (iii) modular integration with inundation surrogates to support rainfall-to-pipe-to-surface digital twins.

We do not quite understand the reviewer's point that using rainfall "introduces other issues". Our framework encodes hydrologic heterogeneity. Static catchment features are explicitly included as node attributes and propagated through the graph, so the model learns rainfall—runoff responses conditioned on local characteristics. This eliminates the need for an external hydrologic model and calibration, increasing portability rather than reducing it.

To make this distinction clear, we will add: (a) expand the related work and methodology sections to clearly contrast runoff-driven surrogates with our end-to-end formulation; and (b) explicitly describe the static hydrologic features included in the model. These clarifications reinforce that our contribution is not a minor variation but an end-to-end surrogate aimed at broader, real-world applications.

2) Use of a heterogeneous GNN: despite using two different types of nodes, i) there is no clear indication of what changes from the GNN perspective other than having different input values, ii) there is no comparison with a baseline to justify the need of heterogeneous nodes, and iii) there is no analysis showing that this representation "enables structured hydrologic representation".

The reviewer questions the value of introducing heterogeneous node types and suggests there is "no clear indication" of what changes beyond input values. This characterization overlooks both the structural and functional importance of the design. Our heterogeneous formulation is not a superficial modification; it enforces a separation between hydrologic and hydraulic processes that a homogeneous architecture cannot capture.

From a modeling standpoint, using distinct node types with distinct feature spaces prevents information dilution. In a homogeneous network, hydrologic descriptors (imperviousness, slope, storage) would have to be artificially assigned to all nodes, forcing meaningless zero-padding for routing-only junctions and outfalls. These non-physical inputs would be propagated through message passing, contaminating the learned functions and blurring the distinction between runoff generation and hydraulic routing. The heterogeneous structure avoids this by restricting hydrologic features to subcatchment nodes, yielding a cleaner and more physically coherent learning task. This is precisely what we mean by "structured hydrologic representation": the graph is designed to mirror the underlying processes, not just the topology.

From an application perspective, the heterogenous design aligns with how stormwater systems are represented in established models such as SWMM, where only a subset of nodes are linked to

subcatchments. This matters for decision-support use cases. For example, when evaluating LID strategies, changing imperviousness or storage should affect only the relevant subcatchment nodes, not be smeared across the entire graph. A homogeneous architecture would treat these localized interventions as global noise, undermining the model's utility. Our approach ensures that physical integrity and local interpretability are preserved.

To underscore this point, we will revise both the introduction and methodology sections to make this rationale explicit, clearly framing the heterogeneous formulation as a design that reflects the dual hydrologic–hydraulic processes and preserves localized interpretability for decision-support applications.

3) Physics-guided constraints: penalizing negative values doesn't enforce/constrain them to be positive. There is also no ablation on whether this component actually improves the model's performance.

The central physics-guided element in our framework is the differential consistency loss, which enforces relational hydraulic constraints by preserving depth and inflow differences across connected nodes. This embeds the physics of flow routing directly into the training objective and is particularly well suited to an end-to-end rainfall-to-hydraulics surrogate. This relational constraint goes beyond the localized feasibility checks used in prior GNN-based studies and represents the distinctive physics-guided contribution of our work.

Alongside this, we include a soft penalty on negative values. It is correct that our penalty does not enforce positivity, but that is intentional: we wanted the penalty to serve as a gentle guide toward feasible outputs while maintaining smooth gradients and stable optimization. In contrast, hard enforcement strategies, such as clamping or ReLU activations, guarantee non-negativity but introduce abrupt, non-differentiable operations that disrupt gradient flow, often leading to the well-known dying ReLU problem and destabilizing learning (Lu et al., 2020). Our formulation integrates seamlessly into the loss function, providing stability without sacrificing physical plausibility.

To strengthen clarity, we will revise the manuscript to (a) clearly distinguish between the differential consistency loss (the main physics-guided component) and the soft penalty, (b) explain why soft guidance was chosen over hard enforcement, and (c) provide a comparison showing performance with and without these terms. Together, these clarifications will underscore that our physics-guided formulation is not incidental, but a principled and impactful design choice.

4) Autoregressive forecasting structure: this same approach has been presented in previous papers (e.g., Bentivoglio et al. 2023, Garzon et al. 2024).

Where our contribution lies is in how the autoregressive structure is integrated into our broader framework. In particular, coupling autoregression with our stability-oriented training strategy enables multi-step prediction within an end-to-end rainfall-to-hydraulics surrogate. This design ensures that errors do not accumulate unchecked over long horizons, which is essential for practical deployment in stormwater applications.

To make this clear, we will update the manuscript to frame autoregression not as a standalone novelty but as a necessary component of a stable, application-ready architecture.

5) Pushforward trick: the version presented here is equivalent in terms of equations to the autoregressive approaches mentioned above. The pushforward trick, as described in Brandstetter et al. 2022, is implemented in a different way from the one here in this paper. Even if it was, there is again no analysis on whether this component benefits the training procedure.

We recognize we did not fully describe key implementation details, which created the impression that our approach is equivalent to prior work. Our pushforward method is an adversarial strategy designed to address distribution shift by training the model on its own realistically flawed predictions. At each step, we introduce these predictions back into the sequence and cut the backpropagation gradient from the initial prediction, which forces the model to learn robustness and error recovery rather than simply minimizing one-step errors. This property, often referred to as zero-stability, is central to achieving reliable long-horizon performance. In our implementation, the stability loss is applied across three time steps, which provides stronger regularization and error recovery capacity than a single-step formulation.

This design shares the underlying logic of the pushforward trick introduced by Brandstetter et al. (2022), but our implementation extends it by applying the loss across multiple steps rather than only the immediate next step. That distinction makes the method more effective for stormwater applications where multi-step stability is critical.

It is also important to clarify how our approach differs from curriculum learning strategies. Curriculum methods typically extend the prediction horizon during training, with gradients propagated through the entire trajectory to minimize accumulated error. By contrast, our pushforward strategy introduces perturbations at each step while blocking gradients from flowing through the initial prediction. This ensures the model learns to recover from errors rather than simply avoid them. This principle is fundamentally different from the multi-step-ahead loss of Bentivoglio et al. (2023), which propagates gradients through all time steps, as their own description confirms.

In our revision, we will (a) describe the implementation of the pushforward trick with sufficient technical detail to avoid ambiguity, and (b) include results comparing training with and without the pushforward component to demonstrate its contribution to stability.

One of the novelties of the paper but is not mentioned here is the estimation of flood volumes directly from the predicted hydraulic variables, but it does not justify the paper being published on its own.

We thank the reviewer for this constructive comment. We will highlight this point as we thoroughly and carefully revise and clarify the contributions of our work.

I think that the paper would have to go through a large series of modifications for it to be novel enough to justify its publication.

Because of these concerns, I have to recommend rejection.

**General comments:**

Introduction:

You mention as main knowledge gap the use of runoff rather than rainfall as a node input.

Despite it being true that you could gain a bit more speed-up from emulating that part as well, you also end up in a new challenge where your model has now to generalize also over different hydrological parameters that can be simply disregarded as an input for the GNN model otherwise.

I didn't see in the experiments any results on how your model would behave for changes in the hydrological characteristics of the node catchments.

This limitation seems to be missing as well from the model limitations later on.

Moreover, there is a paper from Garzon et al. (2024b) that already includes edge-level features.

You should at least compare how you models differ and clarify that there are already examples tackling this gap in the introduction.

We respectfully note that our contribution is not limited to "taking rainfall as an input" but to developing a fully an end-to-end, rainfall-driven surrogate. As explained in our response to your Comment 1 above, this represents a fundamentally different problem formulation from runoff-driven surrogates. Prior models emulate only the hydraulic routing once runoff is pre-computed; our framework directly couples rainfall, hydrologic characteristics, and hydraulic states in a single surrogate.

A key aspect of our design is the inclusion of hydrological parameters (imperviousness, slope, soil type, catchment area, etc.) as node features. This is intentional: these descriptors allow the network to learn how rainfall interacts with physical catchment properties to produce runoff. By encoding this information directly, the model can condition rainfall—runoff transformation on local characteristics, which is essential for eventual transferability across systems with different hydrologic settings. Ignoring these descriptors might reduce inputs, but it would come at the expense of physical realism and the ability to generalize beyond a single study area.

The reviewer suggests that using rainfall inputs introduces a new challenge for transferability, but this issue is not unique to our framework. Any surrogate, whether rainfall-driven or runoff-driven, must be retrained or adapted when applied to a new catchment, since both rely on catchment-specific attributes. Within the study area used in our study, these parameters (e.g., imperviousness, soil type, catchment area) are fixed and do not vary across rainfall events, so variability cannot be tested in this dataset. Their importance emerges only when transferring to new study areas, which is a natural next step beyond this proof-of-concept study. We will revise the manuscript to make this clear and to emphasize that transferability is a broader challenge for surrogate modeling, not a limitation introduced by the end-to-end design.

Finally, with respect to Garzon et al. (2024b), we agree that the introduction can better situate our study in relation to existing work. We will revise the introduction to clarify how our framework differs in scope (end-to-end rainfall-to-hydraulics rather than runoff-to-hydraulics), structure (heterogeneous node design), and application value (supporting decision-relevant tasks such as LID optimization and integration with inundation surrogates).

**Section 2:**

I don't see the point in having a background section as it currently is.

Consider removing it and integrating it directly in the methodology, since there are already some overlaps in the GNN part and the part on SWMM does not seem to be relevant for the rest of the paper.

We understand the reviewer's perspective on the structure of the background section. While we believe it is useful to provide context on both GNNs and SWMM, we agree that this material can be streamlined. In the revision, we will integrate the essential details directly into the methodology and remove overlaps to improve flow and conciseness.

**Section 3:**

**Figure 1:**

The whole figure needs to be re-designed as it is quite confusing in the current state.

For instance, x 1, x 4, etc. are not defined anywhere in the figure;

What should be a vector  $\text{textbf}\{x\}$  1 is written as a scalar;

These vectors (e.g.,  $x_1$ ) seem to have the same input repeated ( $x_1^1$  present twice, same in the others), though I suppose it has different features;

The message-passing block could be better referenced to the "GN Block" you have above "Aggregation";

You define both  $hat\{y\}_i$  and  $hat\{y\}_i$  which seems to indicate the same variable, even if you are referring to two different outputs;

From the figure, it also seems like the decoded node features are given as input to the edge decoder: is it the case? This was not clear in the rest of the paper.

The colouring of the cells inside the figure may seem to help showing that the features are mixing but it also create more confusion, especially after the first mixing.

There is a variable called h\_{selected} that appears only in the figure and is unclear.

The caption of the figure should also better explain what is happening in the figure, clarifying the different variables.

I would recommend a simpler design, removing all MLP figures, colored shapes, and case-specific names (x\_1 or x\_14). If you want to leave the latter, consider adding a reference graph with the corresponding node and edge names.

Thank you for the detailed and constructive feedback on Figure 1. We will completely redesign the figure to address all of your comments and improve its clarity. The new design will be a simplified schematic that uses generic, consistent notation, such as clarifying vector components with proper boldface type, and removes confusing elements.

You noted the notation  $\hat{y}_i$  and  $\hat{y}_{ij}$  was unclear; our intent was to define two distinct outputs for node-level and edge-level predictions, respectively. The revised figure will make this separation visually explicit with clearer labeling. Regarding the decoder, you asked if decoded node features are used as input to the edge decoder. This is not the case, and we will make this clearer in the revised figure. Our model conceptually uses separate decoding paths: final node embeddings are mapped to node predictions, and final edge embeddings are mapped to edge predictions. The redesigned figure will clearly illustrate these two distinct pathways to accurately represent the model's architecture.

line 181: you start defining static and dynamic features without having them introduced before.

You also include xy coordinates as inputs: can you expand more on the implications on transferability of this approach to other case studies? (similarly for example to what is done in Garzon et al. 2024)

We will revise Section 3.1 to introduce and clearly define the two categories of input features (static vs. dynamic) before listing them.

Regarding the use of x-y coordinates: these are included to help the network capture spatial relationships and hydraulic gradients between connected nodes, not to tie the model to a specific coordinate system. In practice, the inputs for any new case study would be derived from system schematics (pipe geometry, node elevations, subcatchment properties) and organized in a SWMM-compatible format that preserves connectivity. The x-y information therefore acts as a relational feature that enhances learning, rather than a barrier to transferability.

That said, rigorous testing of cross-watershed transferability was beyond the scope of this proof-of-concept study. We will explicitly acknowledge this limitation in the revised manuscript and expand the discussion to highlight what would be needed for generalization to other systems, such as multi-catchment training across diverse hydrologic and spatial settings.

line 186: is there a reason why you chose 3 time steps as a input history? How is this related to "support multi-step autoregressive prediction"? You can predict autoregressively even without multiple input time steps in theory.

Yes, the autoregression can in principle be performed with a single historical step. However, in practice, the choice of input window size involves a trade-off between providing enough temporal context and avoiding overfitting on long histories. Prior work (Pfaff et al., 2021, Sanchez-Gonzalez et al., 2020) shows that small windows of 2–3 steps are effective for capturing short-term dynamics without introducing spurious long-range correlations.

Our approach aligns with this principle. A single input step provides only the instantaneous state of the system. For stormwater dynamics, this is insufficient because runoff generation and hydraulic response depend not just on current rainfall but also on short-term accumulation and recent flow conditions. By providing three historical steps, the model can capture this trend and momentum, which are essential for correctly representing how rainfall translates into runoff and routing. At the same time, limiting the window to three avoids the risk of overfitting that arises with longer histories.

Our choice of a three-time-step window therefore serves two purposes. First, it provides the minimal temporal context needed to capture accumulation effects and short-term system momentum, which cannot be represented with a single input step. Second, it aligns with our pushforward training strategy: the model predicts at time t using the true, observed data from t-2 and t-1, then shifts into autoregressive mode where t+1, t+2, and t+3 are generated recursively from its own, previously generated outputs. As illustrated in Figure 4-b, this method deliberately forces the model to learn from its own "noisy" predictions while gradients are detached from the initial step, which stabilizes optimization. The three-step choice is therefore not arbitrary but reflects both physical and methodological considerations.

We will revise the manuscript to make this rationale explicit so the connection between input window length, physical dynamics, and the pushforward strategy is clear.

line 185: how do you calculate the node inflow? is it determined by the predicted edge flows?

The dynamic input features for our model are sourced from SWMM simulations across diverse storm events. Among these features, 'inflow' refers to the Total Inflow, which represents the combined flow into a node from surface runoff (Lateral Inflow), and all connected upstream conduits. We will revise the manuscript to clarify this point.

line 194: with "globally normalized" do you mean that you create a single scaler for all variables or do you have one for each variable?

By "globally normalized" we mean that each variable (e.g., junction depth, conduit flow) is normalized separately using statistics computed across all events in the dataset. This ensures comparability across events while removing scale disparities and stabilizing the learning process. We will revise the manuscript to clarify this point.

Figure 2 and line 198:

It seems that your model predicts in one go the following three time steps, but in the figure's caption it seems like you first predict t+1, then t+2, and so on. Which of the two is it? And if it's the first, why, again, choosing 3 as a number of time steps?

Figure 3 seems to clarify this issue as it shows that you do the second. Please change the rest of the paper clarifying that your model predicts only one step into the future, meaning that it's not limited to only 3 steps ahead.

Since your model can predict any number of future time steps, why do you limit your predictions to the same size as the input time steps?

In other works, the two characteristics are independent (e.g., Bentivoglio et al. 2023, Garzon et al. 2024).

Yes, our model is a one-step-ahead predictor that is applied recursively to generate multi-step forecasts. Figure 3 correctly illustrates this autoregressive process. We agree that this point was not made clearly enough in the manuscript, and we will revise the methodology to state explicitly that the model predicts one step at a time, not multiple steps in a single forward pass.

The confusion in Figure 2 arises from how the pushforward training strategy was visualized. The three-step window shown there reflects the training configuration used to implement the pushforward trick, not the model's inherent prediction capability. We will revise the figures and captions so they consistently illustrate the autoregressive structure and avoid this ambiguity.

Regarding the prediction horizon: in our implementation, the number of past input steps and the forecast horizon are independent hyperparameters. In practice, we found a horizon of three steps yielded the best validation performance during hyperparameter tuning, but the framework is not restricted to this configuration. The model is flexible to accommodate different input histories and forecast horizons depending on the application.

line 197: maybe add a reference to Fig. 2, otherwise it was unclear to me how you combine static and dynamic features.

Thanks for your comment. We will clarify more on that and also reference Figure 2.

**Figure 3:**

Part a is could be a bit clearer: you can for example clarify what are the inputs and outputs since so far they might look the same.

You should also better show which inputs are taken from ground-truth simulations and which ones are predicted by your model, as the update in the red box seems to show that you use ground-truth data as input.

This figure seems to also indicate that there are no overlaps between training windows from t-p to t+p. This decreases by a factor p the number of training samples, making the training faster but potentially less effective. Please add some justification for this choice.

Part b: same comment regarding the coloring of ground-truth data as before.

Thank you for your detailed feedback on Figure 3. We will make sure to clarify the figures in the revised version to make this process unambiguous.

Regarding the training windows, our training methodology does employ overlapping windows. We use a sliding window approach to generate training samples, which maximizes the use of the available simulation data. We will explicitly clarify this in the manuscript and ensure the revised figures accurately reflect this process.

**Section 3.2:**

While it is true that there are no hydraulic paper that consider the pushforward trick, there are other papers that you cite that deal with the same problem using directly a multi-step-ahead loss that generalizes the pushforward trick to multiple time steps ahead (Bentivoglio et al. 2023, Garzon et al. 2024).

Indeed, Eq. 9 is identical to that of Bentivoglio et al. 2023 and Garzon et al. 2024, so please clarify your novelty claims.

Moreover, in the original paper from Brandstetter et al. 2022, the gradients were cut after the first time step, but it seems you are not doing that.

The reviewer is correct that Brandstetter et al. (2022) cut gradients after the first step, and our implementation follows the same principle. During the autoregressive rollout in training, gradients are detached after the initial prediction so that the model is trained to recover from its own imperfect outputs rather than simply backpropagating through a perfect trajectory. This was not made explicit in the manuscript, and we will revise the methods section to state this clearly.

While Equation 9 is formally identical to the loss functions in Bentivoglio et al. (2023) and Garzon et al. (2024), it is crucial to distinguish between their multi-step-ahead loss and our pushforward approach. A multi-step loss (e.g., Bentivoglio et al., 2023; Garzón et al., 2024) propagates gradients through the entire horizon to minimize accumulated error, teaching trajectory perfection. The pushforward trick instead introduces distribution shift deliberately by feeding back noisy predictions while cutting gradients after the first step, thereby enforcing robustness and error recovery (zero-stability). Although both aim to address long-horizon stability, they rely on different principles.

We will revise the manuscript to make these distinctions explicit so that the novelty of our formulation is clear.

Eq 9 and 10: you are missing the underscript v on both variables y. Also mention at some point that these are mean squared errors.

The reviewer is correct that the subscripts were missing; we will add the "\_v" to the y variables in both equations and explicitly state that they represent mean squared errors.

Eq. 10: if you are always predicting your outputs based on ground-truth data, is this equivalent to a one-step-ahead loss accumulated over multiple time steps?

The review is correct that Equation 10 can be interpreted as a one-step-ahead loss accumulated over multiple steps. We believe this question arises because such a formulation resembles curriculum or multi-step loss accumulation strategies used in prior work. Our framework, however, differs in how the two loss terms are structured and how gradients are handled.

Real loss ( $L_{real}$ , Fig. 4a): At each step, the model predicts the next state using ground-truth observations as inputs (e.g., depth and inflow at t-1 and t to predict t+1). This is repeated recursively for three future steps (t+1, t+2, t+3). At each step, the prediction is compared directly against the corresponding ground-truth state, and the mean squared errors across all three steps are averaged to define  $L_{real}$ . This ensures the network learns to reproduce the true dynamics under perfect input conditions.

Stability loss ( $L_{stability}$ , Fig. 4b): In this phase, we deliberately introduce noisy inputs to mimic inference-time conditions. First, the model uses observed states at t-2 and t-1 to predict the state at t. After this first step, the model's own predictions are recursively fed back as inputs: for example, the pair consisting of t-1 and the predicted  $t(t-1,\hat{t})$  is used to predict t+1; then, the pair consisting of  $(\hat{t},\hat{t}+1)$ , both model predictions, is used to predict t+2; and so on. At each rollout step, the predicted state is compared directly against the corresponding ground-truth state, and the mean squared errors across the rollout horizon are averaged to define  $L_{Stability}$ . Importantly, gradients are detached after the first step, so the model does not backpropagate through the entire trajectory. This forces the model to learn how to recover from its own imperfect outputs, rather than only minimizing accumulated error through a perfect rollout.

Together, these two losses balance accuracy and robustness.  $L_{real}$  ensures the model aligns closely with true system dynamics under ideal inputs, while  $L_{stability}$  explicitly prepares it for autoregressive deployment by training it to handle noisy, imperfect inputs. This gradient-handling strategy distinguishes our pushforward approach from prior multi-step loss accumulation methods (e.g., Bentivoglio et al., 2023), which propagate gradients through the entire horizon to enforce trajectory perfection rather than error recovery.

We will revise the manuscript to clarify this distinction so that the complementary roles of  $L_{real}$  and  $L_{stability}$  are unambiguous.

**Figure 4:**

As for the previous figures, it would be to have a legend that clarifies what each color represents, mainly to highlight which outputs are predicted and which ones are ground-truth.

Also, the top figure includes 3 previous input time steps while figure a and b only 2.

It might make the figure easier to understand if you compressed all static and dynamic features that are always ground-truth into a single block of a more transparent shade.

We appreciate this detailed feedback. As clarified in earlier responses, our model consistently uses a two-step input history for its predictions. The discrepancy in Figure 4 reflects the visualization of the pushforward training setup rather than the model's actual prediction process. The model begins with two observed inputs (t-2 and t-1) to predict the current state at time t, and then proceeds autoregressively, feeding its own predictions back as inputs.

In the revision, we will update Figure 4 to eliminate this ambiguity. Specifically, we will (a) add a clear legend distinguishing predicted vs. ground-truth values, (b) correct the input history to consistently reflect the two-step design, and (c) compress static and ground-truth dynamic features into a single block to simplify interpretation.

**Section 3.3:**

This section and 3.2 should be merged for clarity as they both deal with a loss function.

We will revise the manuscript to merge the two sections.

line 239: penalty term: penalising negative values doesn't "ensure that your model respects hydraulic feasibility", it just helps skewing the results to that direction.

We agree with the reviewer's observation that the penalty functions as a soft constraint, guiding predictions toward physical plausibility rather than strictly enforcing feasibility. As noted in our response to Major Comment 3, this was a deliberate design choice to preserve smooth optimization while still biasing the model toward feasible outputs. In the revision, we will update the text to replace "ensure" with a more accurate term such as "promote" or "encourage," so that the manuscript reflects this behavior precisely.

Some other works, like Palmitessa et al. (2022), directly use a ReLU activation to guarantee that there are no negative values. Did you also try out this approach? Does the presence of this loss term improve the results?

We did not implement the hard constraint approach described by Palitessa et al. (2022). This guarantees feasibility, but applying a similar hard clipping (e.g., ReLU) directly to network outputs has a significant drawback: whenever a prediction is negative, the output is forced to zero and the gradient vanishes. The model therefore receives no corrective signal, making it unable to learn how to avoid producing non-physical predictions. In contrast, our penalty loss acts as a soft constraint. It provides a constant, non-zero gradient whenever the prediction is negative, which continuously nudges the network toward feasible values while still allowing the optimization process to converge smoothly. We opted for this approach because it provides a more stable and informative learning signal, which is especially important in an end-to-end rainfall-to-hydraulics surrogate where robustness depends on recursive predictions.

We will clarify this rationale in the revised manuscript so the distinction between hard and soft enforcement strategies is explicit.

line 247: please use the same notation on flows, inflows, etc. throughout the paper. I think using these symbols (also in the rest of Section 3) makes it clearer to identify which variables you are considering.

We will revise the manuscript to ensure consistent notation.

Eq 13: If you decide to keep the penalty term, with a valid justification, please define it before mentioning it in the loss function.

We will add the following formal mathematical definition for the penalty loss,  $L_{penalty}$ , before it appears in the total loss function:

$$L_{penalty} = \lambda_d \sum_{i \in V} ReLU(-\hat{d}_i) + \lambda_c \sum_{(i,j) \in E} ReLU(-\hat{d}_{ij})$$

where V is the set of nodes, E is the set of edges,  $\hat{d}_i$  is the predicted depth at node i, is the predicted depth in conduit (i, j),  $\lambda_d$  and  $\lambda_c$  are weighting coefficients controlling the penalty strength, and  $ReLU(\cdot)$  is the Rectified Linear Unit function, which penalizes only non-physical negative depth predictions.

line 250: you seem to imply that in validation the loss is given by the base term, which comprises both ground-truth and predicted inputs.

Is this the case or are you only considering the "stability" term?

Yes, that is correct. During validation, we compute the base loss, which combines the Real Loss (calculated with ground-truth inputs) and the Stability Loss (calculated with predicted inputs). We will revise the manuscript to make this point explicit so there is no ambiguity.

line 260: why do you also measure the Pearson correlation?

We included the Pearson correlation (*r*) because it measures an aspect of performance that is not captured by RMSE or NSE. While RMSE quantifies absolute error and NSE evaluates the model's accuracy relative to the observed mean, Pearson's r specifically measures the trend agreement and timing of the predictions relative to the observations.

Consider, for example, a case where the model correctly tracks the rise and fall of a flood hydrograph but systematically over- or underestimates flow magnitudes. In such a scenario, RMSE and NSE would penalize the model heavily for magnitude errors, even though the temporal dynamics are captured very well. Pearson's r, by contrast, would still be close to 1, reflecting that the model successfully reproduces the timing and shape of the hydrograph. This highlights the added value of including r, since it provides insight into temporal agreement that cannot be disentangled from magnitude errors using RMSE or NSE alone.

This distinction is also illustrated in Figure 15, where certain locations show high correlation (r) even when NSE is low, emphasizing how r complements RMSE and NSE in evaluating model performance.

**Section 4:**

line 285: why did you consider this coastal case study if you then have to adapt the real conditions (pipes with sea water) to a simplified version? Doesn't the model work with presence of water in the system?

The coastal case study was selected because it provided a realistic and complex network that could serve as a high-fidelity testbed. In this initial study, we deliberately simplified the SWMM simulation by excluding tidal backflow effects so that the focus remained on the core research challenge: demonstrating that a GNN can learn both rainfall-runoff generation and network routing directly from rainfall inputs.

This simplification should not be interpreted as a limitation of the GNN's ability. The GNN learns the dynamics that are represented in its training data. Extending the framework to include boundary conditions such as tidal influence is straightforward once the end-to-end rainfall-to-hydraulics

surrogate has been established. We will clarify this rationale in the manuscript and note explicitly that incorporating tidal forcing is an important direction for future work.

**Figure 5:**

It would help to have an elevation map as well to visualize the slope of the sewer system.

We will modify Figure 5 to include an elevation layer to help visualize the system's topography and slope.

It seems that all SWMM nodes are flooded according to Waze. How did you calibrate the SWMM model then based on these observations (lines 288-289)?

To calibrate SWMM against Waze flood observations, we iteratively adjusted pipe sizes in tidally influenced areas using elevation data, achieving a reasonable match with reported flooded nodes. Although the real system frequently floods due to combined rainfall and tide, this baseline model (of resized pipes) was intended as a simplified yet representative test case for GNN development.

It also seems like there are some disconnected parts in your system. Is it an error in the map or do you model separate parts?

Yes, this is an error in the map, and we will correct and modify it in the revised version.

lines 291-293: What does this sensitivity anlaysis mean? Is this the variability of the static catchment attributes across all SWMM nodes?

Sensitivity analysis of static catchment attributes (e.g., subcatchment width) confirmed that modest perturbations did not meaningfully alter model outcomes (flooded nodes), which is important given the ungauged nature of the watershed and reliance on binary flood validation via Waze. Finally, only individual rainfall events were simulated, so no consecutive or antecedent rainfall conditions were considered.

line 298: "each event was scaled by factors of 1.2, 1.4, or 1.6." How did you choose which ones to scale with which factor? The final number of events is 300 but you start with 85 events.

Our goal in scaling rainfall events was to enrich the dataset with more intense rainfall scenarios and to expand the sample size to 300 events. We began with 85 original events. To create the remaining 215 augmented events, we applied multiplicative scaling factors of 1.2, 1.4, and 1.6. From the full set of 255 possible augmentations (85 events × 3 scaling factors), we randomly selected 215 unique combinations of (event, scaling factor). The final dataset therefore consists of 85 original events plus 215 augmented events.

Moreover, it would be useful to show the variability in hyetographs between training, valiation, and testing, as, based on the basically perfect results, it looks like there might be some overlap between them.

We will add a new figure or table to the manuscript to address this. It will compare the statistical properties of the hyetographs across the training and testing sets to demonstrate their variability and confirm the independence of the test set.

lines 367-372: these are a repetition of the training details that to me mostly add confusion, as it seemed that now your model was predicting only 3 steps ahead into the future (line 368), while in line 373 it seems that you don't use any ground-truth. Please clarify in case it's the latter.

You are correct that the text as written blurs two separate procedures. We will revise the manuscript to make this distinction explicit.

The first paragraph (lines 367-372) describes the training process. The model is trained with a sliding-window approach. It repeatedly receives a 3-step history of ground-truth data and is trained to predict the next 3 ground-truth steps. This teaches the network short-term dynamics under ideal input conditions.

The second paragraph (starting line 373) explains the inference (or testing) phase. Once trained, the model is initialized with the first 3 observed time steps. From there, it operates autoregressively, predicting one step at a time and feeding its own outputs back as inputs. This process can be continued for any forecast horizon. In inference, ground-truth hydraulic states are not used beyond initialization; rainfall remains the only external input at each future step.

Figure 3 depicts this distinction, and we will revise the text to more explicitly separate training from inference so there is no ambiguity about how predictions are generated.

In the results section, do you consider dry-period events for the simulations? If so, can you tell how much they affect the NSE values?

We did not explicitly separate wet and dry periods during training or evaluation. Instead, the model was trained on sliding windows with shuffled mini-batches, ensuring that both low-flow and active periods were naturally included. Evaluation was conducted over full events using standard autoregressive rollout.

For context, the regime-splitting approach referenced in prior work relies on a 1 mm rainfall depth threshold over a 9-step window, with rebalanced sampling between wet and dry regimes. Our history length is shorter (3 steps rather than 9), so such explicit stratification was not necessary in our setup.

For clarity, we will note this in the revised manuscript and emphasize that the reported NSE values reflect model performance across entire events, without regime-specific stratification.

**Section 5.3:**

Why do you test the flooding performance only on a single test scenario?

The flood performance analysis focused on a single, high-intensity storm, which we selected as a representative event to rigorously test the model's flood detection capabilities under challenging conditions. We note, however, that results for multiple rainfall events are already included in other sections of the manuscript. For the sake of brevity, we initially presented only one scenario here, but in the revised version we can add complementary results summarizing flood detection metrics (CSI, precision, and recall) across several additional storm events from our test set to demonstrate that the model's strong performance is consistent.

**Figure 13:**

It should include a comparison with the ground-truth flooded nodes according to SWMM, which you compare against.

We will modify Figure 13 to include a direct comparison with the ground-truth flooded nodes from the SWMM simulation.

lines 468-469: please add a reference for this claim.

We will revise the manuscript to support this claim with references from the broader GNN literature (e.g., Horie and Mitsume, 2022).

**Section 5.3.2:**

This section on limitations doesn't address one important limitation that you introduce with your approach, i.e., the range of hydrological parameters that you now should be modelling in place of SWMM.

It would be also useful to add some insights on the transferability of this model to other case studies. This is an important observation, and we will expand the limitations section to discuss it explicitly.

In a stormwater system, hydrological parameters (e.g., subcatchment area, slope, imperviousness) are fixed once the model is calibrated and do not vary across rainfall events within the same catchment. Therefore, in the present study, these parameters remain constant across simulations.

Their importance emerges when considering transferability to other catchments, where hydrologic properties differ. However, this issue is not unique to an end-to-end surrogate. Even runoff-driven surrogates that emulate the hydraulic routing component are not directly transferable without retraining on new catchment data. Transferability is therefore a broader challenge for data-driven surrogates not a limitation created by our end-to-end formulation.

We will revise the manuscript to make this point clear and to emphasize that improving generalizability requires training on data spanning multiple catchments with diverse hydrological parameters. Such a strategy would enable the model to capture transferable relationships between catchment properties and hydraulic response, rather than fitting a single system in isolation.

**Section 5:**

At some point of the paper, you should point out the computational times needed for the model to train and test, since one of the main drivers of your research is speed.

We will add a new subsection to the "Results" section detailing the computational performance of our model. This will include the total training time, the average inference time per storm, and a direct speed-up factor compared to the SWMM model.

**Section 6:**

Please remove the whole section as it does not add any relevant information to the paper.

It also resembles a lot the interactive dashboard provided by Garzon et al. (2024) (https://github.com/alextremo0205/SWMM GNN Repository Paper version).

We will remove Section 6 from the manuscript.

**Other comments:**

line 95: there is a reference error

Thank you for raising this issue. We will fix this in the revised manuscript.

line 382: please reference fig 10 and then 11, and not viceversa.

Thank you for raising this issue. We will fix this in the revised manuscript.

**Reference:**

Lu, L., Shin, Y., Su, Y., & Karniadakis, G. E. (2019). Dying relu and initialization: Theory and numerical examples. *arXiv preprint arXiv:1903.06733*.

Pfaff, T., Fortunato, M., Sanchez-Gonzalez, A., & Battaglia, P. (2020, October). Learning mesh-based simulation with graph networks. In *International conference on learning representations*.

Sanchez-Gonzalez, A., Godwin, J., Pfaff, T., Ying, R., Leskovec, J., & Battaglia, P. (2020, November). Learning to simulate complex physics with graph networks. In *International conference on machine learning* (pp. 8459-8468). PMLR.

Garzón, A., Kapelan, Z., Langeveld, J., & Taormina, R. (2024). Transferable and data efficient metamodeling of storm water system nodal depths using auto-regressive graph neural networks. Water Research, 266, 122396.

Bentivoglio, R., Isufi, E., Jonkman, S.N. and Taormina, R., 2023. Rapid spatio-temporal flood modelling via hydraulics-based graph neural networks. Hydrology and Earth System Sciences, 27(23), pp.4227-4246.

Brandstetter, J., Worrall, D. and Welling, M., 2022. Message passing neural PDE solvers. arXiv preprint arXiv:2202.03376.

Horie, M., & Mitsume, N. (2022). Physics-embedded neural networks: Graph neural pde solvers with mixed boundary conditions. *Advances in Neural Information Processing Systems*, *35*, 23218-23229.

---

## Author Comment (AC3)

1. This paper is concerned with the development of a hydrological-hydraulic surrogate architecture for urban drainage systems. The paper is within scope of the journal. It also presents some interesting, novel ideas that go beyond previous work. However, these are not at all supported by empirical results. Instead, all results are presented for the new framework as is. We therefore don't know at all if the study actually made progress compared to previous similar works. In addition, the paper is not always concise, and the methodological details are occasionally incomplete and sometimes confusing.

I have provided details below. I don't think that these issues can be addressed in a normal revision, and I therefore suggest to reject and invite the resubmission of a thoroughly revised manuscript that also includes a number of new results supporting the methodological progress.

We thank the reviewer for carefully evaluating our work and for the constructive and detailed feedback. We agree that additional empirical analyses, including ablation studies, component-level comparisons, and computational performance benchmarks, will further strengthen the manuscript by isolating the contribution of individual architectural choices. These analyses will help illustrate methodological progress relative to previous studies, beyond evaluating the unified framework as a whole.

At the same time, we would like to clarify that the manuscript already includes substantial empirical evidence demonstrating that the proposed unified surrogate performs well across a range of hydrological–hydraulic tasks. The present results evaluate predictive accuracy for node states, conduit flows, flood-node detection, and flood-volume estimation across multiple storm events. These findings support the central proof-of-concept contribution: that a single end-to-end GNN can successfully learn the coupled rainfall–runoff and flow-routing process. The additional analyses we will add in response to the reviewer's suggestions are intended to complement this existing evidence by providing finer-grained insight into the role of each model component, rather than to fill an absence of empirical support.

We also acknowledge the reviewer's observation that certain methodological sections can be made more concise and clearer. In the revised manuscript, we will streamline the presentation, improve figure design, and expand explanations where needed to ensure that the methodological pipeline is transparent and easy to follow. Many of the points raised, e.g., clarifying feature definitions, refining figure annotations, and specifying training configurations, are straightforward improvements in presentation, and we will address each of them carefully in the revision.

**Detailed comments:**

2. l64: I disagree on this point. The main reason why previous studies focused on the hydraulics is because that is where most of the computation time is used. Hydrological models are already fast and can also be integrated in differentiable form into a surrogate framework.

There also aren't any results for what we gain from including the hydrological model in the surrogate architecture, neither in terms of accuracy nor speed.

We agree that hydraulic routing is typically the dominant computational bottleneck in physics-based models . Our intention was not to suggest that hydrological models are computationally slow on their own, but rather the value of learning the coupled hydrological-hydraulic process within a single unified surrogate. In this formulation, the model directly maps rainfall and catchment attributes to node depths, conduit flows, and flooding behavior, instead of relying on a separate runoff-generation step followed by a distinct hydraulic module.

We recognize that hydrological models can be embedded in differentiable form within surrogate frameworks, but such designs generally retain a modular structure in which precipitation is first transformed to runoff through a predefined hydrological component and then routed hydraulically. By contrast, our GNN-based framework replaces the entire simulation pipeline with an end-to-end mapping that learns rainfall–runoff–routing interactions jointly. The inclusion of hydrological parameters (e.g., imperviousness, slope) as node features allows the network to infer how rainfall interacts with local catchment characteristics and how these effects propagate through the drainage graph, rather than assuming a fixed hydrological mapping upstream of the surrogate.

Empirically, Sections 5.2 and 5.3 already show that this unified surrogate reproduces key hydrological–hydraulic behaviors across multiple events, including node depths, conduit flows, flood node detection, and flood volume estimation. This provides the core proof-of-concept for the study, demonstrating that the proposed model *can* learn the complete coupled rainfall–runoff–routing process, which is a non-trivial advancement beyond modular or hydraulics-only surrogate designs.

To further clarify the benefits of this formulation, we will include a new subsection in the Results section reporting the model's computational performance. Because the goal of our approach is to replace the entire physics-based pipeline, the most meaningful metric is the end-to-end speedup relative to the full SWMM workflow rather than a component-wise hydrology comparison, which would not reflect the different problem formulations. The new subsection will therefore quantify the overall pipeline acceleration achieved by the unified GNN surrogate and clarify its computational advantages.

We also note that the unified, end-to-end formulation enables modeling capabilities that modular or decoupled surrogates cannot easily support. By learning the full rainfall–runoff–routing process, the model can act as a fast 1D surrogate for coupled 1D/2D flood simulation frameworks, where its predicted nodal flood volumes can serve as source terms for a 2D inundation surrogate. Additionally, because the model learns how subcatchment properties (e.g., imperviousness, slope) affect hydraulic response, it can support decision-relevant scenario analyses such as evaluating the hydraulic impact of various LID strategies. We will add a brief clarification of these potential applications in the Discussion section to make these broader implications explicit.

3. l95: bad reference

Thank you. We will fix this in the revised manuscript.

4. Section 2.1: do we need to describe SWMM? maybe this can be moved to an appendix

In the revised version, we will move the description of SWMM to an Appendix.

5. Section 2.2.1: This section can also be shortened. Do mention that this is a graph convolution? Why was it selected over GineConv and GATConv that seem to have performed better in other studies?

Thank you for the constructive feedback. We agree that this section can be streamlined, and we will revise it accordingly. We will also make it explicit that the "GN block" used in our processor is a form of graph convolution based on the Graph Network (GN) framework and implements a message-passing structure.

Regarding the choice of this architecture over GATConv or GINEConv: our primary requirement is the ability to jointly learn dynamic edge states (conduit flows) and dynamic node states (junction

depths) within the same model. This requires an operator that explicitly updates edge embeddings as part of the iterative message-passing process. The standard GATConv does not natively incorporate edge features and is designed primarily for node-level tasks, requiring additional custom fusion layers to embed edge information (Veličković et al., 2018, Zhang et al. 2024). GINEConv does incorporate edge features but does so through a more restrictive additive formulation and does not update edge states iteratively during message passing (Garzón et al. (2024a, Garzón et al., 2024b).

In contrast, the GN Block we use, which is based on architectures explicitly designed for physics simulation (Sanchez-Gonzalez et al., 2018; Pfaff et al., 2021), performs a two-stage update that first computes new edge embeddings via a learned non-linear function of the edge and its two incident nodes, and then updates node states using these dynamically updated edges. This architecture is specifically designed for domains where relational (edge) dynamics are as important as object (node) dynamics, making it well suited for stormwater hydraulics.

We will revise Section 2.2.1 to shorten the description and clearly justify this architectural choice so that the rationale is transparent to the reader.

6. Figure 1: This figure is both overloaded and can at the same time not be understood from the figure and the caption alone. The lower part illustrating the message passing process is mostly confusing, and can safely be removed, because this process is already well documented in the text as well as the literature.

In the upper part, it is not clear what node types a and b are, what the indices of x1, x4, x14 and x13 indicate, if four different encoder MLPs are used or if the same MLP is used, that we start working on edges in the processor and symbolise inputs for three different edges - each composed of upstream and downstream node properties as well as edge properties, they values are being aggregated over edges connected to each node, that we need to apply another MLP to decode, and how we finally arrive at predictions for t+2 and t+3. The figure just repeats the message passing step, so it may well be a consideration to drop it.

Thank you for the constructive feedback. We will completely redesign the figure in the revised version. The new version will be a simplified schematic that directly addresses the issues raised by the reviewer: we will remove the redundant lower panel illustrating message passing, clearly define 'Node type a' and 'Node type b' , replace the confusing indices (e.g., x1, x14) with a general data flow, clarify the encoder/decoder steps, and be corrected to clearly illustrate a single-step autoregressive prediction ($t \rightarrow t + 1$) to resolve the inconsistency the reviewer noted.

7. Figure 2: Text in the top part of the figure is too small. Information in this subfigure and the table is repeated, which is confusing. It is not clear to me why went nice coordinates would be necessary inputs to the model. The network is fully defined from adjacency and edge length (and slope, which is missing). dx and dy are not explained.

Thank you for the constructive feedback. We will revise this figure to address the issues raised by the reviewer.

Regarding the feature inputs, we acknowledge the inconsistency in listing "slope" as a conduit feature and will correct this in the revision. In the revised version, we will also clarify the role of the spatial features. Absolute node coordinates (x, y) are provided as static node attributes, and dx and dy are derived edge features representing the relative horizontal displacement between connected nodes. Together with node elevations, these features allow the model to infer conduit

orientation and slope implicitly, while also providing spatial context that improves learning of location-dependent hydraulic behavior. We will clearly define dx and dy and revise both the figure and the accompanying text to ensure this rationale is transparent.

8. Eq. 6: Why do you use difference in depth and not hydraulic head, which is the actual driving force behind the flow through edges?

We agree that the hydraulic head gradient ($\Delta H$), not just the water depth difference ($\Delta y$), is the true physical driver of flow. In our formulation, the model receives both components needed to infer $\Delta H$: the water depth difference ($\Delta y$) as the Diff-depth feature, and the elevation difference ($\Delta z$) is available implicitly because node elevations ($z_i \ and \ z_j$) are included in the node features used during message passing (Eq. 3). Therefore, the GNN has access to $\Delta H = \Delta z + \Delta y = (z_i - z_j) + (y_i - y_j)$, and can learn the corresponding hydraulic head gradient along each edge.

We will revise Section 3.1 to make this justification clearer and to explicitly state how the model reconstructs the hydraulic head gradient from the supplied features.

9. l200: I don't understand this. In Figure 2 you illustrate that you predict three steps at the time. Here and in Figure 3 it looks like you move one step at the time. Please clarify and make sure that the description is consistent throughout.

Thank you for pointing out this inconsistency. Our model is one-step-ahead predictor that is applied recursively (autoregressively) to generate multi-step forecasts. Figure 3 correctly illustrates this autoregressive structure. We agree that this was not explained clearly enough in the manuscript, and we will revise the methodology to explicitly state that the model predicts one step at a time rather than multiple steps in a single forward pass.

The confusion in Figure 2 arises from how the pushforward training strategy was visualized. The three-step window shown there reflects the training configuration used for the pushforward trick, not the model's inherent prediction mechanism. We will revise the figures, their captions, and the corresponding text so both figures consistently represent the autoregressive nature of the model and avoid ambiguity.

10. Section 3.2: The works of Palmitessa, Garzon and Bentivoglio (for 2D flooding) all condition the model on its own predictions during training. What you do different is that you include an additional loss term where the model is conditioned on the ground truth. You need to show that this improves training stability compared to the situation where we condition on model predictions only. This may actually be the case, because conditioning on the model predictions essentially corresponds to a recurrent network, which are well known for vanishing gradient issues. In any case, the argument is currently formulated the wrong way round, and results for this aspect are missing. Figure 4 is again overcomplicating things. I'm able to understand the concepts from the text, but not from the figure, which is overloaded with information that is not relevant in this context.

Thank you for this helpful observation. We agree that additional empirical evidence isolating the effect of our stability loss would strengthen the manuscript. We will include an ablation study comparing training with and without the stability component to demonstrate its contribution to stability and error recovery.

We also appreciate the reviewer's comparison to prior work. Our approach differs from standard autoregressive conditioning or multi-step losses used in previous 2D flood surrogates. Specifically,

our contribution is more than simply adding a loss term conditioned on the ground truth ($L_{real}$). The novelty lies in the specific implementation of our stability loss ($L_{stability}$), which uses a gradient-blocking pushforward strategy. During training, the model's own predictions are reintroduced into the input sequence, but the gradient from the initial prediction is cut. This differs fundamentally from standard autoregressive or multi-step losses, where gradients propagate through the entire predicted trajectory. By blocking the gradient pathway, $L_{stability}$ forces the network to learn robustness to distribution shift (i.e., to recover from its own imperfect predictions) rather than optimizing only short-term accuracy. We extend this mechanism by applying the stability loss over a three-step window (n = 3), which we found provides stronger regularization for stormwater applications. This gradient-blocking, multi-step stability design is distinct from the training formulations in previous studies, and we will revise the manuscript to describe this implementation more clearly.

To avoid confusion, we agree that Figure 4 currently contains more information than needed. In the revision, we will simplify the figure and revise the explanation of the training strategy so that the distinction between our stability loss and conventional autoregressive conditioning is clear.

11. Section 3.3: These loss functions are interesting and potentially valuable, but you don't illustrate that you actually achieve better training performance than previous studies that have just used L_real. How do we know that these modifications actually make a difference?

Thank you for this helpful comment. We agree that the manuscript should explicitly demonstrate the contribution of the physics-guided loss components. In the revised version, we will include a new ablation study comparing

- a baseline model trained with $L_{base}$ only
- variants trained with $L_{base} + L_{penalty}$ and $L_{base} + L_{diff}$, and
- the full model trained with all loss terms

This ablation will quantify the effect of each component on training stability and predictive accuracy. We will also revise Section 3.3 to clarify the motivation for these loss terms and how they complement $L_{real}$ in guiding physically consistent predictions.

12. Section 5.1: It seems all the information in the table is also described here. Please remove either text or table. How many forecast steps did you use for training? How much GPU memory did you have available?

Thank you for the helpful suggestion. We will remove the redundant text in Section 5.1 and refer directly to Table 1 to streamline the presentation. For clarity, the training forecast horizon used in the pushforward strategy was three steps, and the model was trained on an NVIDIA RTX 4090 GPU with 24 GB of VRAM. We will include these details explicitly in the revised manuscript.

13. Figure 12: Do the boxplots illustrate variations of average NSE value across events, or do they combine variations of NSE on node and event level, i.e. is 0.89 the lowest average NSE for all events, or is it the lowest value observed for any node in any event?

Thank you for the question. The boxplots do not represent variations of average NSE *across events*. Instead, each boxplot summarizes the per-entity performance (i.e., per-node or per-conduit), where the NSE values are computed for each entity over all test events combined. Thus, values such as 0.89 represent the lowest per-entity NSE observed across all events, not the lowest average eventlevel NSE. We will revise the caption and accompanying text for Figure 12 to clearly describe this aggregation method.

14. Section 5.3: I think this assessment should be done as an average across events with surcharges. Considering only a single event is very unreliable.

The flood performance analysis focused on a single, high-intensity storm, which we selected as a representative event to rigorously test the model's flood detection capabilities under challenging conditions. We note, however, that results for multiple rainfall events are already included in other sections of the manuscript. For the sake of brevity, we initially presented only one scenario here, but in the revised version we can add complementary results summarizing flood detection metrics (CSI, precision, and recall) across several additional storm events from our test set to demonstrate that the model's strong performance is consistent.

15. Figure 14: axis labels miss units

Thank you for raising this issue. We will fix this in the revised manuscript.

16. Results for computational speed up are missing entirely

Thank you for pointing this out. We agree that including computational performance is important. In the revised manuscript, we will add a dedicated subsection in the Results section reporting the end-to-end computational speed of the unified surrogate relative to the full SWMM workflow. This subsection will quantify runtime and overall pipeline speedup, including descriptive statistics comparing CPU-based and GPU-based runs, to clearly demonstrate the computational benefits of the proposed model.

17. Section 5.3.2: The behaviour in Fig. 15 is already clear from earlier results. This figure can therefore be moved to an appendix. It is also not common practice to include figures in the discussion.

Thank you for the suggestion. Figure 15 was included in Section 5.3.2 to visually support the discussion on model limitations, but we agree that it can be moved to the appendix. In the revised manuscript, we will move Figure 15 to an appendix and adjust the text in Section 5.3.2 accordingly.

18. In addition, this section is missing an important limitation, namely that the trained model remains system specific.

Thank you for pointing out this important limitation. We agree that the trained model in its current form is system-specific to the Haven Creek Watershed and does not generalizable to a new domain without retraining. We will add this explicitly to the limitation section.

At the same time, we would like to clarify that the proposed framework itself is not inherently system-specific. The unified, end-to-end surrogate formulation is general and can be trained on other stormwater systems or extended to additional domains. This paper serves as a proof-of-concept for the methodology, and we will revise the manuscript to clearly distinguish between the generality of the framework and the system-specific nature of the trained model used in this study.

19. Section 6: this section does not belong into a scientific paper

We will remove Section 6 from the manuscript as suggested.

**References**

- Garzón, A., Kapelan, Z., Langeveld, J., & Taormina, R. (2024a). Transferable and data efficient metamodeling of storm water system nodal depths using auto-regressive graph neural networks. *Water Research, 266*, 122396.

- Garzón, A., Kapelan, Z., Langeveld, J., & Taormina, R. (2024b). Accelerating Urban Drainage Simulations: A Data-Efficient GNN Metamodel for SWMM Flowrates. *Engineering Proceedings, 69*(1), 137.

- Pfaff, T., Fortunato, M., Sanchez-Gonzalez, A., & Battaglia, P. W. (2021). Learning mesh-based simulation with graph networks. *International Conference on Learning Representations (ICLR)*.

- Sanchez-Gonzalez, A., Heess, N., Springenberg, J. T., Merel, J., Riedmiller, M., Hadsell, R., & Battaglia,P. (2018). Graph networks as learnable physics engines for inference and control. *International Conference on Machine Learning (ICML)*.

- Veličković, P., Cucurull, G., Casanova, A., Romero, A., Liò, P., & Bengio, Y. (2018). Graph attention networks. *International Conference on Learning Representations (ICLR)*.

- Zhang, Z., Tian, W., Lu, C., Liao, Z., & Yuan, Z. (2024). Graph neural network-based surrogate modelling for real-time hydraulic prediction of urban drainage networks. *Water Research, 263*, 122142.